# ConvFormer: Revisiting Token Mixers for Sequential User Modeling

## Abstract

Sequential user modeling is essential for building recommender systems, aiming to predict users' subsequent preferences based on their historical behavior. Despite the widespread success of the Transformer architecture in various domains, we observe that its self-attentive token mixer is outperformed by simpler strategies in the realm of sequential user modeling. This observation motivates our study, which aims to revisit and optimize the design of token mixers for this specific field. We start by examining the core building blocks of the self-attentive token mixer, identifying three empirically-validated criteria essential for designing effective token mixers in sequential user models. To validate the utility of these criteria, we develop ConvFormer, a streamlined modification to the Transformer architecture that satisfies the proposed criteria simultaneously. We also present an acceleration technique to handle the computational cost of processing long sequences. Experimental results on four public datasets reveal that even a simple model, when designed in accordance with the proposed criteria, can surpass various complex and delicate solutions, validating the efficacy of the proposed criteria.

## 1 Introduction

Recommender system serves as a cornerstone for various online services such as e-commerce (Smith & Linden, 2017), advertising (Zhou et al., 2018), and movie & TV (Gomez-Uribe & Hunt, 2016). In parallel to other tasks such as collaborative filtering (He et al., 2017) and click-through rate prediction (Lian et al., 2018), sequential user modeling, which is typically formulated as a next-item-prediction problem, is a foundational task in crafting effective recommendation engines. The challenge lies in accurately capturing and understanding the evolving patterns of user preferences from sequential behavioral data (Kang & McAuley, 2018) for next item prediction.

Advances in deep learning have spurred the development of sequential user models based on neural networks, represented by recurrent neural networks (RNN) (Hidasi & Karatzoglou, 2018; Ren et al., 2019), convolutional neural networks (CNN) (Tang & Wang, 2018; Yuan et al., 2019), graph neural networks (GNN) (Zhuo et al., 2022), and Transformers (Kang & McAuley, 2018; Sun et al., 2019). Transformer-style models, in particular, have been particularly transformative in diverse fields with domain-specific adoptions, exemplified by Swin Transformer (Liu et al., 2021) for images and AlphaFold-v2 (Jumper et al., 2021) for protein structures. In contrast, current progress in sequential user modeling stays in some direct applications of Transformer structure (Sun et al., 2019) with minimal domain-specific adjustments. This has led to instances where simpler approaches, such as MLP-like and CNN-like modules, outperform the more complex self-attentive token mixers in Transformers (Zhou et al., 2022). Such observations spur our reevaluation of Transformer-like structures, especially self-attentive token mixers, in the realm of sequential user modeling.

The success of self-attentive token mixers is often ascribed to the scalability and flexibility of the *item-to-item* token mixing paradigm. Yet, this paradigm's lack of sensitivity to the order of items – treating them as if order permutations are equivalent – limits its efficacy in scenarios where temporal order is crucial, such as in tracking evolving user preferences. This is corroborated by empirical evidence suggesting that simpler, order-sensitive alternatives, such as MLP layers (Li et al., 2022), learnable filters, pure FFT (Lee-Thorp et al., 2022) and even arbitrary projections (Tay et al., 2020), can yield superior or competitive results, which challenges the indispensability of the item-to-item paradigm in sequential user models and motivate us to mine the core features that support Trans-

former's efficacy. By deconstructing various aspects of self-attentive token mixer, we identify two factors favouring its performance: a large receptive field and a lightweight architecture; in contrast, the item-to-item interaction model may be a detriment rather than an asset.

In light of these insights, we propose three criteria for devising token-mixers in sequential user models: order sensitivity, a large receptive field, and a lightweight architecture. To validate these criteria, we introduce ConvFormer, we propose **ConvFormer**, a simple yet effective adaptation of the Transformer framework that satisfies all proposed criteria. The core of ConvFormer is replacing the self-attentive token mixer with a depth-wise convolution (DWC) layer and enlarging the receptive field aggressively. Adhering to the proposed criteria, even a straightforward model surpasses various delicate models and achieves state-of-the-art performance, thereby attesting to the validity of our proposed criteria. However, the expanded receptive field poses computational challenges for long input sequence, for which which we develop **ConvFormer-F**, an efficient variant utilizing Fourier convolution to achieve significant speedup with minimal accuracy drop.

To summarize our main contributions:

- We provide a context-specific examination of the self-attentive token mixer and identify three key criteria for designing effective token mixers in sequential user models.

- We propose ConvFormer, a simple yet effective update to the standard Transformer, built upon the proposed criteria. Additionally, we introduce an accelerated version using Fourier convolution to efficiently handle extra-long user behavior sequences.

- Through extensive experimentation, we demonstrate ConvFormer's superior performance over existing models, achieving state-of-the-art results. The overall performance comparison and ablation studies serve to validate the efficacy of the proposed criteria.

## 2 PROBLEM STATEMENT

Consider a set of users $\mathcal{U}$ and items $\mathcal{I}$, for a given user $u \in \mathcal{U}$, we define the behavior sequence as $S_u = \{i_{1,u}, \cdots, i_{L,u}\}$, where L denotes the sequence length, each $i_{l,u}$ represents an item with which the user has interacted chronologically. The goal of sequential user modeling is to model $p(i_{L+1}|S_u)$, the likelihood of the next item that the user may interact given behavior sequence.

We consider the case of item retrieval as recommendations, which models user representations based on their behavior sequences $S_u$. Such user representation subsequently serves as a query to retrieve the next likely item through a simple matching function, such as the dot product. An exemplar to build user representations is the self-attentive recommender (SAR) (Kang & McAuley, 2018). Let $\mathbf{R} \in \mathbb{R}^{L \times D}$ be the embedding of $S_u$ with hidden dimension D, SAR employs a self-attentive token mixer (Vaswani et al., 2017) to model contextual information in $\mathbf{R}$ as follow:

$$\mathbf{A} = \text{SA}(\mathbf{R}) = \text{Softmax}\left(\left(\mathbf{R}\mathbf{W}^{(Q)}\right)\left(\mathbf{R}\mathbf{W}^{(K)}\right)^{\top}/\sqrt{D}\right), \tag{1}$$

where $\mathbf{A} \in \mathbb{R}^{L \times L}$ is the item-to-item attention matrix. The tokens within the input embedding sequence are then mixed as $\mathbf{S} = \mathbf{A}(\mathbf{R}\mathbf{W}^{(V)})$. $\mathbf{S}$ is further refined through a feed-forward network (FFN). The combination of self-attention and FFN forms the building block of SAR, which can be stacked for multiple layers for deep fusion. The final representation of the last item in $S_u$ serves as the user representation, serving as the basis for subsequent item retrieval.

## 3 EXAMINING SELF-ATTENTIVE TOKEN MIXER IN USER MODELING

This section aims to dissect the self-attentive token mixer to identify its critical components for effective sequential user modeling. We specifically scrutinize three aspects: the item-to-item attentive paradigm, the receptive field, and the overall lightweight architecture[1].

---

[1]We use the experimental settings in Section 5.1, e.g., setting the embedding size to 64, the maximum sequence length to 50. Experiments are repeated 5 times with different seeds. We only modify the token mixer matrix $\mathbf{A}$ in (1), maintaining learning objectives and tricks identical to SASRec.

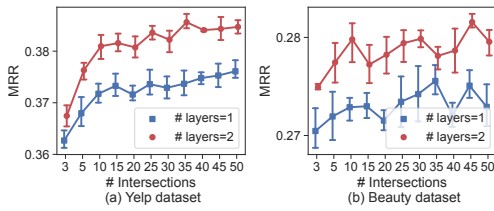 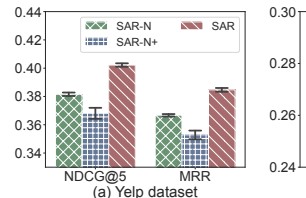 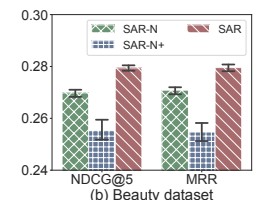

Figure 1: Impact of SAR's receptive field size. Error bar denotes 95% confidence interval.

Figure 2: Impact of SAR's lightweight architecture. Error bar denotes 95% confidence interval.

### 3.1 IS THE ITEM-TO-ITEM TOKEN-MIXER SUITABLE FOR SEQUENTIAL USER MODELING?

The central element in SAR is the item-to-item token mixer $\mathbf{A}$ in (1). To assess its role in sequential user modeling, we replace it with alternative token mixers, resulting in the following variants[2].

- SAR-O (SAR with Order-sensitive weights) utilizes a trainable parameter matrix $\mathbf{A}^{(O)} \in \mathbb{R}^{L \times L}$ that is independent of the input sequence. Unlike SAR which relies solely on position embedding for order information, SAR-O is directly sensitive to the order of the input items. The order sensitivity of token mixer is manifested as: for any $l$-th item, changing the order of other items $l' \neq l$ alters the output representation of the $l$-th item.

- SAR-P (SAR with Personalized weights) modifies SAR-O by using an MLP to dynamically generate attention scores based on the input $\mathbf{R}$ to this block, wherein $\mathbf{A}^{(P)}[l] = \mathrm{MLP}(\mathbf{R}[l])$. It enables the customization of $\mathbf{A}^{(P)}$ based on the input while maintaining order sensitivity.

- SAR-R (SAR with Random and order-sensitive weights) is similar to SAR-O but its attention matrix $\mathbf{A}^{(R)} \in \mathbb{R}^{L \times L}$ is randomly initialized, fixed, and non-trainable.

These simple yet order-sensitive alternatives exhibit little performance drop as per Table 1. Notably, SAR-R competes closely with the original SAR despite its non-trainable matrix, which further emphasizes the importance of order sensitivity in sequential user modeling, since SAR-R's primary distinction from SAR is its order sensitivity nature. Furthermore, dynamic weights contribute marginally (SAR-P versus SAR-O), suggesting that the dynamic and adaptive weights in the item-to-item token mixing paradigm are not indispensable for SAR's superiority.

Table 1: Performance of SAR and variants. "*" indicates the variants outperforming SAR with $p$-value$< 0.01$ on the two samples t-test.

| Dataset | Model | H@5 | H@10 | N@5 | N@10 | MRR |
|---------|-------|------|------|------|------|------|
| Sports | SAR | 0.3442 | 0.4647 | 0.2472 | 0.2861 | 0.2504 |
| | SAR-O | 0.3474* | 0.4682* | 0.2497* | 0.2887* | 0.2526 |
| | SAR-P | **0.3478*** | **0.4686*** | **0.2503*** | **0.2891*** | **0.2531*** |
| | SAR-R | 0.3438 | 0.4646 | 0.2470 | 0.2860 | 0.2503 |
| Yelp | SAR | 0.5684 | 0.7446 | 0.4018 | 0.4590 | 0.3841 |
| | SAR-O | 0.5713 | 0.7472 | 0.4048 | 0.4618* | 0.3870* |
| | SAR-P | **0.5731*** | **0.7473** | **0.4061*** | **0.4626*** | **0.3878*** |
| | SAR-R | 0.5692 | 0.7455 | 0.4033 | 0.4604 | 0.3858 |

Recent studies questioning the necessity of self-attention across different applications (Lee-Thorp et al., 2022; Tolstikhin et al., 2021) support our findings, suggesting that the item-to-item attentive paradigm might be limiting the efficacy of Transformer-style sequential user models by not adequately considering the inherent order of items. Therefore, incorporating architectures that explicitly recognize item order could potentially enhance performance. We extend the analysis to additional datasets and provide further statistical validation in Table A6.

### 3.2 IS THE LARGE RECEPTIVE FIELD ESSENTIAL FOR SEQUENTIAL USER MODELING?

Another distinctive feature of SAR is its large receptive field. It allows each element in a user sequence to interact directly with others within a single self-attention layer, facilitating the efficient capture of long-term user behavior patterns. We hypothesize that this large receptive field is a key contributor to SAR's performance. To test this hypothesis, we modify SAR's attention matrix to consider only the interactions between each item and its K nearest neighbors. This is achieved by

---

[2]We provide graphical illustrations and detailed implementations of these SAR variants in Appendix C.2.

using a window mask $\Gamma(\mathrm{K})$, defined such that $\Gamma_{ij} = 1$ if $|i - j| \leq \mathrm{K}$ and $-\inf$ otherwise for $0 \leq i, j \leq \mathrm{L}$. The attention matrix $\mathbf{A}$ in (1) is multiplied with $\Gamma(\mathrm{K})$ before calculating Softmax.

Our experiments reveal a positive correlation between the receptive field size and SAR's performance, as shown in Figure 1. For instance, increasing K from 3 to 45 leads to a significant improvement in MRR: a 4.5% increase on the Yelp dataset and 2.43% on the Beauty dataset. These results underscore the importance of a large receptive field for SAR's performance.

### 3.3 IS THE LIGHTWEIGHT ARCHITECTURE ESSENTIAL?

Given that a larger receptive field potentially increases model complexity, maintaining a lightweight overall architecture is critical to reduce the risk of over-parameterization. This balance is essential not just as a technical nuance but as a key factor in unlocking the advantages of a large receptive field. For example, SAR exemplifies this principle by sharing query, key, and value mapping parameters $\mathbf{W}^{(*)}$ across all time steps in (1). To empirically validate the importance of such lightweight structure, we introduce two SAR variants:

- SAR with non-shared parameters (SAR-N), where the query, key and value mapping parameters (i.e., $\mathbf{W}^{(*)}$ in (1)) are unique at different time steps.
- SAR with more non-shared parameters (SAR-N+), where all items in the input sequence are concatenated to generate query, key, and value vectors though an MLP module.

Both variants above sacrifice the lightweight of vanilla SAR for order-sensitivity and high capacity. According to Figure 2, they exhibit a performance decline compared to the standard SAR model. Specifically, SAR-N+ shows a relative MRR decrease of 3.11% on Yelp and 8.91% on Beauty, while SAR-N experiences a relative MRR drop of 4.65% on Yelp and 8.27% on Beauty. These results highlight the critical role of a lightweight architecture in mitigating the risks posed by large receptive fields, thus ensuring the continued efficacy of SAR models.

## 4 PROPOSED METHOD

### 4.1 THREE CRITERIA FOR SEQUENTIAL USER MODELING

The empirical studies in Section 3 suggest that a large perception field and a lightweight architecture are key factors contributing to the superior performance of SAR. However, the item-to-item paradigm is identified as a limitation due to its insensitivity to item order. Based on these insights, we propose three design principles for constructing effective token mixers in sequential user modeling:

1. The token-mixer should be sensitive to the order of items, to capture sequential patterns such as evolving preference from user behaviors;

2. The token-mixer should encompass a large receptive field, to capture and exploit long-term patterns in user behavior sequences;

3. The token-mixer should maintain a lightweight architecture, to mitigate the risk of overfitting that may result from a large receptive field.

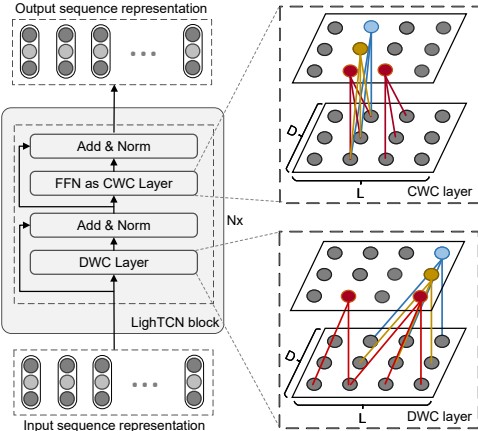

Figure 3: The core structure of ConvFormer.

Prevailing sequential user models satisfy some of these criteria, but fall short in meeting them simultaneously. SAR-based models encompass large receptive field and lightweight architecture, but overlook the order sensitivity (Criterion 1); RNN-based models are order sensitive but typically lack a large receptive field (Criterion 2). CNN-based methods, such as Caser (Tang & Wang, 2018), employ narrow receptive field and conventional convolution operators, overlooking the potential of using large receptive field and light convolution operators (Criteria 2 and 3). These gaps highlight the need for a new architecture that aligns with our proposed criteria for performance improvement.

### 4.2 THE CONVFORMER ARCHITECTURE

In light of these, we develop ConvFormer, a simple yet effective update to the standard SAR model. The primary technical contribution is to replace the item-to-item mechanism with a novel LighTCN layer, which involves a large receptive field and ensures item order sensitivity while maintaining a lightweight overall architecture, thereby satisfying all three proposed criteria simultaneously. Figure 3 presents the core building block of ConvFormer, and the workflow is described below.

#### 4.2.1 EMBEDDING LAYER

ConvFormer starts with an embedding layer, which converts high-dimensional, one-hot item indices into a dense, lower-dimensional representation. This process involves an item embedding lookup table $\mathbf{E}^{(\mathrm{I})} \in \mathbb{R}^{\mathcal{I} \times \mathrm{D}}$ and a learnable position encoding matrix $\mathbf{E}^{(\mathrm{P})} \in \mathbb{R}^{\mathrm{L} \times \mathrm{D}}$. Specifically, a user's historical behaviors is represented as

$$\hat{\mathbf{E}} = [\mathbf{E}^{(\mathrm{I})}_{i_1} + \mathbf{E}^{(\mathrm{P})}_1, \mathbf{E}^{(\mathrm{I})}_{i_2} + \mathbf{E}^{(\mathrm{P})}_2, ..., \mathbf{E}^{(\mathrm{I})}_{i_{\mathrm{L}}} + \mathbf{E}^{(\mathrm{P})}_{\mathrm{L}}], \tag{2}$$

where $\mathbf{E}^{(\mathrm{I})}_{i_j}$ is the item embedding of the $j$-th item in $S_u$. If the sequence length is less than L, pad zeros on the left side. To avoid overfitting and ensure a stable training process, following Kang & McAuley (2018) and Zhou et al. (2022), we refine (2) with dropout and layer normalization:

$$\hat{\mathbf{E}} = \mathrm{Dropout}(\mathrm{LayerNorm}(\mathbf{E}^{(\mathrm{I})} + \mathbf{E}^{(\mathrm{P})})). \tag{3}$$

#### 4.2.2 LIGHT TEMPORAL CONVOLUTION NEURAL (LIGHTCN) LAYER

The user embedding is derived by stacking multiple LighTCN layers following the embedding layer. Each LighTCN layer consists of a depth-wise convolution (DWC) layer and a channel-wise convolution (CWC) layer. The DWC layer operates on each channel of the input independently, considering each dimension of the embedding vector as a separate channel. Specifically, for a given layer, let $\mathbf{R} \in \mathbb{R}^{\mathrm{L} \times \mathrm{D}}$ be the representation of input sequence (for the first layer, $\mathbf{R} = \hat{\mathbf{E}}$), $\mathbf{C} \in \mathbb{R}^{\mathrm{K} \times \mathrm{D}}$ be the convolution kernel with size K, we conduct depth-wise convolution along the temporal axis:

$$\mathrm{DWC}(\mathbf{R})_{l,d} = \mathrm{Pad}(\sum_{k=1}^{\mathrm{K}} \mathbf{R}_{l+k-1,d} * \mathbf{C}_{k,d}), \quad d = 1, \ldots, \mathrm{D}, \tag{4}$$

where the output is left-padded to ensure $\mathrm{DWC}(\mathbf{R}) \in \mathbb{R}^{\mathrm{L} \times \mathrm{D}}$. Following Kang & McAuley (2018); Zhou et al. (2022), we incorporate skip connections, layer normalization, and dropout operations:

$$\hat{\mathbf{R}} = \mathrm{LayerNorm}(\mathbf{R} + \mathrm{Dropout}(\mathrm{DWC}(\mathbf{R}))). \tag{5}$$

While the DWC layer captures linear temporal characteristics on each individual channel, it ignores non-linear interactions and channel-wise dependencies. Thus, we integrate a CWC layer, analogous to the Feed-Forward Network (FFN) layer in Transformers (Vaswani et al., 2017). This layer employs a $1 \times 1$ convolution defined as $f(\mathbf{x}) = \mathbf{x}\mathbf{W} + \mathbf{b}$ and a ReLU activation function. At each time step, the CWC layer operates as follows:

$$\mathrm{CWC}(\hat{\mathbf{R}}_l) = \mathrm{FFN}(\hat{\mathbf{R}}_l) = f(\mathrm{ReLU}(f(\hat{\mathbf{R}}_l))), \quad l = 1, 2, \ldots, \mathrm{L}, \tag{6}$$

which is similarly refined by the skip connection, dropout and layer normalization technologies:

$$\tilde{\mathbf{R}} = \mathrm{LayerNorm}(\hat{\mathbf{R}} + \mathrm{Dropout}(\mathrm{CWC}(\hat{\mathbf{R}}))). \tag{7}$$

#### 4.2.3 DOT-PRODUCT SCORER

ConvFormer utilizes a dot-product scorer for next-item prediction. Let $\mathbf{e}_c$ be the embedding of an item $c$ within the item embedding matrix $\mathbf{E}^{(\mathrm{I})}$, $\bar{\mathbf{R}}$ be the output from the final LighTCN layer. Adhering to the two-tower retrieval paradigm (Kang & McAuley, 2018; Zhou et al., 2022), the output representation at the last step, $\bar{\mathbf{R}}$, is employed as the user's representation. We then estimate the likelihood of a user interacting with item $c$ at the L + 1 step as $p(i_{\mathrm{L}+1} = c|i_{1:\mathrm{L}}) = \mathrm{sigmoid}(\mathbf{e}_c^\top \bar{\mathbf{R}}[\mathrm{L}])$.

The training process involves updating the learnable weights of the model to minimize a ranking loss function, which is identical to Kang & McAuley (2018) as follow:

$$\mathcal{L} = -\sum_{u \in \mathcal{U}} \sum_{l=1}^{L} \log(p(i_{l+1}|i_{1:l})) - \log(1 - p(i_{l+1}^-|i_{1:l})), \tag{8}$$

where each ground-truth item $i_{l+1}$ is paired with a negative item $i_{l+1}^-$ that is randomly sampled. The training sequences are generated in an autoregressive manner, which mirrors the function of causal mask in Transformer models to prevent the model from using future information in predicting.

### 4.3 ACCELERATED APPROXIMATION ALGORITHM

One potential concern with LighTCN is the computational cost associated with a large receptive field, particularly when dealing with lengthy behavior sequences. As the receptive field extends to encompass the entire sequence length, the complexity can escalate to $\mathcal{O}(L^2)$, on par with the complexity of SASRec. As a strategic component to handle such complexity, we introduce the Fourier convolution technique (Mathieu et al., 2014) and construct an accelerated version of ConvFormer, denoted as **ConvFormer-F**. The rationale comes from the convolution theorem (Oppenheim et al., 2001): the convolution in the temporal domain is equivalent to a Hadamard product in the Fourier domain, which yields a more efficient computation of the DWC layer:

$$\text{DWC}(\mathbf{R}) = \mathcal{F}^{-1}\left(\mathcal{F}(\mathbf{R}) \odot \mathcal{F}(\mathbf{C})\right) \tag{9}$$

where $\odot$ indicates the Hadamard point-wise product, $\mathbf{C}$ is right-padded with zeros to match the length of $\mathbf{R}$. The Fast Fourier Transform $\mathcal{F}$ (Oppenheim et al., 2001) reduces the computational complexity from approximate $\mathcal{O}(L^2)$ to $\mathcal{O}(L \log(L))$, making it advantageous for processing extremely lengthy sequences. We offer a detailed computational workflow and a comparative analysis of accuracy and speed in Appendix B. Overall, Fourier convolution can effectively handle the complexity of ConvFormer in handling lengthy user behavior sequences without loss of accuracy.

## 5 EXPERIMENTS

To demonstrate the efficacy of both the proposed criteria and ConvFormer, which is a simple yet inspiring model built upon these criteria, the five aspects as follows deserve empirical investigation.

- **Performance:** *Does ConvFormer work?* We compare ConvFormer against state-of-the-art baselines, with the 1-vs-99 performance in Table 2 and the full-sort performance in Appendix A.1.

- **Gains:** *Why does it work?* We deconstruct various aspects of ConvFormer in Section 5.3 to identify the sources of its accuracy gain and back up the efficacy of the proposed three criteria. We provide additional comparisons and rigorous statistical tests in Appendix A.4 and A.5.

- **Generality:** *Does it work in other datasets and tasks?* We investigate the performance on a large industrial dataset in Appendix A.2, and a general CTR prediction task in Appendix A.3.

- **Speed:** *Does ConvFormer-F reduces running time while preserving accuracy?* We compare actual running time of SASRec, ConvFormer and ConvFormer-F in various settings in Appendix B.

### 5.1 EXPERIMENTAL SETUP

**Dataset.** We select four public sequential user modeling datasets adhering to Zhou et al. (2022). We organize each user's interactions chronologically, using the latest item for testing, the second to last item for validation, and the remaining items for training. Users or items with less than five interactions are excluded. The processed datasets' statistics are summarized in Table A3.

**Evaluation Protocol.** We employ three ranking metrics for evaluation: Top-$k$ hit ratio (H@$k$), Top-$k$ normalized discounted cumulative gain (N@$k$), and mean reciprocal rank (MRR). As for the negative item candidates, we experiment on both types of settings: (1) ranking the positive item against 99 randomly selected non-interacted items for each user; and (2) the full-sort test set where the positive item is ranked alongside all non-interacted items.

Table 2: Performance comparison on four datasets. The bold and underlined fonts indicate the best and second-best performance, respectively. "*" and "**" mark the metrics where ConvFormer outperforms the best baselines with p-value < 0.05 and 0.001, respectively, in the one-sample t-test.

| Dataset | Metric | PopRec | FM | AutoInt | GRU4Rec | Caser | HGN | CLEA | SASRec | BERT4Rec | SRGNN | GCSAN | FMLP-Rec | ConvFormer |
|---------|--------|--------|----|---------|---------|-------|-----|------|--------|----------|-------|-------|----------|------------|
| Beauty | H@1 | 0.0678 | 0.0405 | 0.0447 | 0.1337 | 0.1337 | 0.1683 | 0.1325 | 0.1870 | 0.1531 | 0.1729 | 0.1973 | _0.2011_ | **0.2019** |
| | H@5 | 0.2105 | 0.1461 | 0.1705 | 0.3125 | 0.3032 | 0.3544 | 0.3305 | 0.3741 | 0.3640 | 0.3518 | 0.3678 | _0.4025_ | **0.4119**\*\* |
| | N@5 | 0.1391 | 0.0934 | 0.1063 | 0.2268 | 0.2219 | 0.2656 | 0.2353 | 0.2848 | 0.2622 | 0.2660 | 0.2864 | _0.3070_ | **0.3125**\*\* |
| | H@10 | 0.3386 | 0.2311 | 0.2872 | 0.4106 | 0.3942 | 0.4503 | 0.4426 | 0.4696 | 0.4739 | 0.4484 | 0.4542 | _0.4998_ | **0.5105**\*\* |
| | N@10 | 0.1803 | 0.1207 | 0.1440 | 0.2584 | 0.2512 | 0.2965 | 0.2715 | 0.3156 | 0.2975 | 0.2971 | 0.3143 | _0.3385_ | **0.3443**\*\* |
| | MRR | 0.1558 | 0.1096 | 0.1226 | 0.2308 | 0.2263 | 0.2669 | 0.2376 | 0.2852 | 0.2614 | 0.2686 | 0.2882 | _0.3051_ | **0.3093**\* |
| Sports | H@1 | 0.0763 | 0.0489 | 0.0644 | 0.1160 | 0.1135 | 0.1428 | 0.1114 | 0.1455 | 0.1255 | 0.1419 | _0.1669_ | 0.1646 | **0.1671** |
| | H@5 | 0.2293 | 0.1603 | 0.1982 | 0.3055 | 0.2866 | 0.3349 | 0.3041 | 0.3466 | 0.3375 | 0.3367 | 0.3588 | _0.3803_ | **0.3891**\*\* |
| | N@5 | 0.1538 | 0.1048 | 0.1316 | 0.2126 | 0.2020 | 0.2420 | 0.2096 | 0.2497 | 0.2341 | 0.2418 | 0.2658 | _0.2760_ | **0.2819**\*\* |
| | H@10 | 0.3423 | 0.2491 | 0.2967 | 0.4299 | 0.4014 | 0.4551 | 0.4274 | 0.4622 | 0.4722 | 0.4545 | 0.4737 | _0.5059_ | **0.5116**\*\* |
| | N@10 | 0.1902 | 0.1334 | 0.1633 | 0.2527 | 0.2390 | 0.2806 | 0.2493 | 0.2869 | 0.2775 | 0.2799 | 0.3029 | _0.3165_ | **0.3215**\*\* |
| | MRR | 0.1660 | 0.1202 | 0.1435 | 0.2191 | 0.2100 | 0.2469 | 0.2156 | 0.2520 | 0.2378 | 0.2461 | 0.2691 | _0.2763_ | **0.2808**\*\* |
| Toys | H@1 | 0.0585 | 0.0257 | 0.0448 | 0.0997 | 0.1114 | 0.1504 | 0.1104 | 0.1878 | 0.1262 | 0.1600 | _0.1996_ | 0.1935 | **0.2007** |
| | H@5 | 0.1977 | 0.0978 | 0.1471 | 0.2795 | 0.2614 | 0.3276 | 0.3055 | 0.3682 | 0.3344 | 0.3389 | 0.3613 | **0.4063** | _0.4033_ |
| | N@5 | 0.1286 | 0.0614 | 0.0960 | 0.1919 | 0.1885 | 0.2423 | 0.2102 | 0.2820 | 0.2327 | 0.2528 | 0.2836 | _0.3046_ | **0.3069**\* |
| | H@10 | 0.3008 | 0.1715 | 0.2369 | 0.3896 | 0.3540 | 0.4211 | 0.4207 | 0.4663 | 0.4493 | 0.4413 | 0.4509 | _0.5062_ | **0.5100** |
| | N@10 | 0.1618 | 0.0850 | 0.1248 | 0.2274 | 0.2183 | 0.2724 | 0.2473 | 0.3136 | 0.2698 | 0.2857 | 0.3125 | _0.3368_ | **0.3384**\* |
| | MRR | 0.1430 | 0.0819 | 0.1131 | 0.1973 | 0.1967 | 0.2454 | 0.2138 | 0.2842 | 0.2338 | 0.2566 | 0.2871 | _0.3012_ | **0.3048**\* |
| Yelp | H@1 | 0.0801 | 0.0624 | 0.0731 | 0.2053 | 0.2188 | 0.2428 | 0.2102 | 0.2375 | 0.2405 | 0.2176 | 0.2493 | _0.2727_ | **0.2816**\*\* |
| | H@5 | 0.2415 | 0.2036 | 0.2249 | 0.5437 | 0.5111 | 0.5768 | 0.5707 | 0.5745 | 0.5976 | 0.5442 | 0.5725 | _0.6191_ | **0.6347**\*\* |
| | N@5 | 0.1622 | 0.1333 | 0.1501 | 0.3784 | 0.3696 | 0.4162 | 0.3955 | 0.4113 | 0.4252 | 0.3860 | 0.4162 | _0.4527_ | **0.4653**\*\* |
| | H@10 | 0.3609 | 0.3153 | 0.3367 | 0.7265 | 0.6661 | 0.7411 | 0.7473 | 0.7373 | 0.7597 | 0.7096 | 0.7371 | _0.7720_ | **0.7863**\*\* |
| | N@10 | 0.2007 | 0.1692 | 0.1860 | 0.4375 | 0.4198 | 0.4695 | 0.4527 | 0.4642 | 0.4778 | 0.4395 | 0.4696 | _0.5024_ | **0.5146**\*\* |
| | MRR | 0.1740 | 0.1470 | 0.1616 | 0.3630 | 0.3595 | 0.3988 | 0.3751 | 0.3927 | 0.4026 | 0.3711 | 0.4006 | _0.4299_ | **0.4406**\*\* |

**Baseline Models.** The collection of baselines includes[3]: (1) **PopRec**, **FM** (Rendle, 2010), and **AutoInt** (Song et al., 2019) are non-sequential models; (2) **GRU4Rec** (Hidasi & Karatzoglou, 2018), **Caser** (Tang & Wang, 2018), **HGN** (Huang et al., 2020), **CLEA** (Qin et al., 2021), and **SRGNN** (Wu et al., 2019) are representative sequential baselines which do not involve Transformer architectures; (3) **SASRec** (Kang & McAuley, 2018), **BERT4Rec** (Sun et al., 2019), **GCSAN** (Xu et al., 2019) and **FMLP-Rec** (Zhou et al., 2022) are baselines that (partially) rely on Transformer architectures.

## 5.2 OVERALL PERFORMANCE

The results using the 1-vs-99 test protocol are reported in Table 2. To summarize our observations:

- Sequential models outperform non-sequential methods such as PopRec, FM and AutoInt, which underscores the importance of item ordering information. SAR-based models like SASRec and GCSAN achieve better performance over RNN-based (*e.g.,* GRU4Rec), CNN-based (*e.g.,* Caser) and GNN-based models (*e.g.,* SRGNN), which can be attributed to their lightweight architecture and large receptive field, aligning with criteria (2) and (3) from Section 4.1. Furthermore, FMLP-Rec outperforms other baseline methods, which could be attributed to the unique sensitivity of its filter layer to item order, substantiating the efficacy of the criterion (1).

- ConvFormer significantly outperforms most baseline models across four datasets, with most differences being statistically significant. In addition, the all-convolution architecture of ConvFormer is both computationally efficient and parallelizable, making it efficient for training and inference, as discussed in Appendix B. Thus, ConvFormer proves to be an effective and efficient solution to sequential user modeling. The superiority of ConvFormer demonstrates that adhering to the criteria proposed, *even a very simple model can outperform many sophisticated solutions and achieve leading performance*, thereby validating the efficacy of the proposed criteria.

## 5.3 ABLATION STUDIES

We have showcased the effectiveness of the three criteria through the superior performance of ConvFormer. In this section, we conduct ablation studies to further assess the role of each criterion.

---

[3]We respect existing benchmark results, following the settings, datasets and baselines (Zhou et al., 2022).

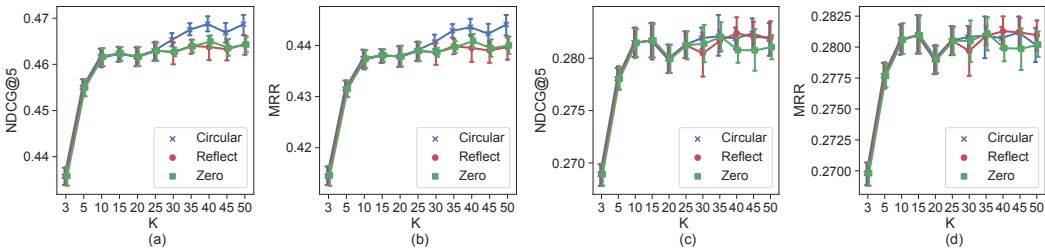

Figure 4: Impact of Convformer's receptive field size K on model performance over Yelp (a-b) and Sports (c-d) datasets. Error bar denotes 95% confidence interval.

### 5.3.1 LARGE RECEPTIVE FIELD

To demonstrate the necessity of a large receptive field, we vary the kernel size K to visualize its impact in Figure 4. Results show that an increase in kernel size leads to improved performance, as evidenced by the rise in MRR from 0.414 at K = 3 to approximately 0.441 at K = 50 on Yelp.

We also investigate the role of padding methods in the convolution operator, with are denoted by *Circular*, *Reflect*, and *Zero* in Figure 4. Specifically, in scenarios with strong behavior periodicity (such as Yelp), circular padding, which preserves the periodic property, performs significantly better than other padding methods. However, in scenarios with weak periodicity in user behaviors (such as Amazon Sports), the performance difference between padding methods is minimal.

Notably, the term *large receptive field* is relative to conventional selection of K such as 3x3 and 7x7, and does not specifically refer to the full receptive field with K = L. Performance gains from enlarging the receptive field have a ceiling; exceeding it offers minimal improvement and introduces challenges like optimization difficulty, increased inference time, and overfitting risk.

### 5.3.2 LIGHTWEIGHT CONVOLUTION

To support the efficacy of lightweight architecture, we replace the LighTCN operator (denoted by Conv-L) of ConvFormer with two variants: the vanilla convolution operator (denoted by Conv-V) and the separable convolution operator (Howard et al., 2017) (denoted by Conv-S). Notably, both Conv-V and Conv-S meet criteria (1) and (2), but violate criterion (3).

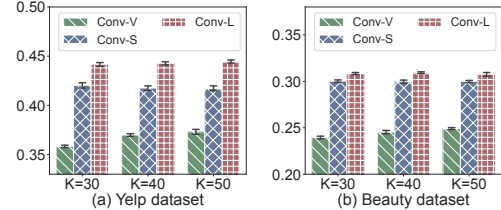

Figure 5: Impact of lightweight convolution.

According to Figure 5, the Conv-L operator (used in our standard ConvFormer) largely outperforms the vanilla convolution operator due to its suppression of over-parameterization. Specifically, when K=30, it improves the MRR by a relative 23.24% on Yelp and 28.71% on Beauty. The Conv-S operator also performs better than the vanilla convolution operator, but worse than Conv-L. Its inferiority is attributed to the extra inter-channel interaction compared to Conv-L. This redundancy increases the risk of over-parameterization, as the subsequent FFN modules are specifically designed for inter-channel interactions.

### 5.3.3 ATTENTION VS. CONVOLUTION

To support the claim that self-attentive modules can be a limitation due to the insensitivity of item order, we replace the LighTCN module with advanced attentive mechanisms. We select to compare ConvFormer with Fastformer (Wu et al., 2021) and PoolingFormer (Zhang et al., 2021), as these two models beat a series of efficient Transformer variants such as LinFormer (Wang et al., 2020) and LongFormer (Beltagy et al., 2020). Notably, both additional baselines satisfy criteria (2) and (3), with large receptive fields and lightweight architectures, but fail to meet criterion (1), i.e., they are developed based on attentive paradigms that is insensitive to item order.

According to Table 3, emerging Transformer variants with large receptive fields and more lightweight architectures could achieve performance gains in sequential user modeling. For example, PoolingFormer improves MRR by approximately 4.2% over SASRec. However, the gap between these methods and ConvFormer remains significant, which suggests that the item-to-item paradigm is a bottleneck for sequential user modeling due to the lack of order sensitivity.

Table 3: Comparison with lightweight attentive methods. "*" marks methods outperforming the best baseline with p-value $< 0.01$.

| Dataset | Model | H@5 | H@10 | N@5 | N@10 | MRR |
|---------|-------|-----|------|-----|------|-----|
| Beauty | FastFormer | 0.3395 | 0.4454 | 0.2438 | 0.2780 | 0.2449 |
| | PoolingFormer | 0.3932 | 0.4925 | 0.2981 | 0.3302 | 0.2971 |
| | ConvFormer | **0.4119*** | **0.5105*** | **0.3125*** | **0.3443*** | **0.3093*** |
| Yelp | FastFormer | 0.5451 | 0.7355 | 0.3727 | 0.4344 | 0.3557 |
| | PoolingFormer | 0.6087 | 0.7663 | 0.4378 | 0.4890 | 0.4144 |
| | ConvFormer | **0.6347*** | **0.7863*** | **0.4653*** | **0.5146*** | **0.4406*** |

## 6 RELATED WORKS

**Sequential user modeling.** The fundamental aspect of sequential user modeling is the handling of user behavior sequences. Consequently, sequential neural architectures like RNN (Hidasi et al., 2016) and CNN (Tang & Wang, 2018; Yuan et al., 2019; 2020) are naturally applicable, often augmented with advanced features such as target-aware attention and memory networks (Zhou et al., 2019; Lian et al., 2021; Wu et al., 2017; Huang et al., 2018). Some scholars posit that a mere sequence of items is insufficient to capture the complexity of user behavior, advocating for the use of Graph Neural Networks (GNNs) to enrich the representational capacity (Wu et al., 2019; Xu et al., 2019; Qiu et al., 2019; Yuan et al., 2020). Currently, the Transformer architecture (Vaswani et al., 2017) has permeated this domain, exemplified by SAR-based approaches (Kang & McAuley, 2018; Sun et al., 2019; Chen et al., 2019; He et al., 2021). However, much of the existing work applies the Transformer indiscriminately, neglecting the unique characteristics of the recommendation field compared to other fields like natural language processing. Recent studies (Li et al., 2022; 2023; Zhou et al., 2022) have indicated that the self-attentive token mixer in Transformer may not be the most effective choice for sequential user modeling, with alternative architectures like all-MLP modules showing promise. These observations serve as the impetus for our investigation into the fundamental criteria for designing token mixers in the context of sequential user modeling.

**Transformer applications and alternatives.** The Transformer architecture, popularized by Devlin et al. (2019), has garnered widespread success in diverse fields, *e.g.,* ViT and Swin-Transformer in vision (Liu et al., 2021; Dosovitskiy et al., 2021) and AlphaFold-v2 in AI4Science (Jumper et al., 2021). These successful applications commonly follow a trajectory of initial direct implementation, subsequently refined by domain-specific adaptations. In parallel, ongoing research aims to explore alternatives to Transformer, particularly the self-attentive token mixer (Lee-Thorp et al., 2022). For example, Tay et al. (2020) replaced the self-attention matrix with a parameterized matrix and achieved performance gains, even positing that a randomly initialized matrix serves as a competitive substitute. Similarly, Tolstikhin et al. (2021) replaces the self-attention layer with multilayer perception, showing advantages over the traditional Transformer in various data-rich domains (Li et al., 2022; 2023). These insights encourage the development of domain-specific token mixers, capitalizing on unique data and task characteristics to optimize performance.

## 7 CONCLUSIONS

In this study, we re-evaluate Transformer-like architectures for sequential user modeling and identify three critical criteria for effective token mixers. Guided by these criteria, we develop ConvFormer, a streamlined modification of the Transformer, augmented with an acceleration technique for computational efficiency. Our findings demonstrate that even a simplified model, when designed in accordance with these criteria, can outperform various complex and delicate solutions, thereby validating the efficacy of the proposed criteria.

**Limitations.** We construct the proposed criteria using the standard two-tower architecture for item retrieval in sequential user modeling. ConvFormer primarily serves as a proof-of-concept, leaving ample scope for future research to explore a broader range of sequential models and tasks to assess the general applicability of our proposed criteria.

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

# A ADDITIONAL EXPERIMENT RESULTS

## A.1 PERFORMANCE ON THE FULL-SORT SETTING

We report the full-sort overall performance in Table A1. The results and main observations are consistent with those on the 1-vs-99 test set in Table 2. For instance, SAR-based models such as SASRec outperform conventional RNN-based models like GRU4Rec and CNN-based models like Caser; ConvFormer demonstrates the best performance among all the methods. The improvements of ConvFormer are more noticeable on the full-sort test set than on the 1-vs-99 test set due to the full-sort setting being more challenging and providing greater opportunities for improvement.

## A.2 PERFORMANCE ON LARGE-SCALE INDUSTRIAL DATASET

We have reported the performance on the public datasets in the main text. However, since these datasets have limited scale, as per Table A3. There is a possibility that the superiority of ConvFormer is due to Transformer's overfitting on these datasets rather than the inefficacy of Transformer's item-to-item paradigm for sequential user modeling. To address this concern, it is crucial to evaluate ConvFormer against competitive baselines, especially Transformer (SASRec), using large-scale industrial datasets.

The results of our experiments on the full-sort test data from this industrial dataset are presented in Table A4 and Table A2. Notably, we observed that GRU4Rec outperforms SASRec on the industrial dataset, which suggests that the evolving of user preference is significant in real-world scenarios, thus highlighting the importance of our criterion (1) regarding order-sensitivity. Moreover, our ConvFormer model achieved a significantly better performance compared to other baselines in particular SASRec. This finding confirms the superiority of ConvFormer and the efficacy of the three criteria we proposed, in large-scale real-world applications.

**Safeguards.** The desensitized and encrypted dataset contains no Personal Identifiable Information (PII). Adequate data protection was carried out during experiment to prevent the risk of data copy leakage. The dataset does not represent any business situation, only used for academic research.

## A.3 PERFORMANCE ON THE GENERAL TASK

In the main text, we have demonstrated the superior performance of ConvFormer in the next-item prediction task using a two-tower retrieval framework. In this section, we aim to highlight the versatility of ConvFormer, specifically the LighTCN layer, as a plug-and-play component that can benefit other general tasks.

We specifically investigate the click-through rate (CTR) estimation task, which involves estimating the CTR and identifying the items that are most likely to be clicked. The CTR estimation task differs from the next-item prediction task in three key aspects: (1) the architecture employed, where CTR estimation typically utilizes a one-tower architecture compared to the two-tower architecture used in next-item prediction; (2) the loss function employed, with CTR estimation using a point-wise loss function as opposed to the pair-wise loss function used in next-item prediction; and (3) the inclusion of user profiles, where CTR estimation involves considering user profiles, unlike next-item prediction which does not require them. Given the significant differences between CTR prediction and next-item prediction tasks, and considering the wide applications of CTR prediction in practical production scenarios, we choose to include CTR prediction as an additional task to evaluate the performance of ConvFormer.

In CTR estimation frameworks, sequential user models serve as interest extractors, and the resulting user representations are concatenated with user profiles and item properties to estimate the CTR. We implement the interest extractors using the LighTCN layer, as well as comparable methods employed in other baseline models. The performance on the Movie-lens dataset is presented in Table A5. To summarize the main observations:

- Replacing the GRU-based interest extractor in DIEN with alternative counterparts shows promise in improving performance in CTR estimation tasks. The relative performance of different models in CTR estimation tasks aligns with their relative performance in next-item prediction tasks.

Table A1: Full-sort performance of different methods on four datasets. The best and second performance methods are marked in bold and underlined fonts, respectively.

| Datasets | Metric | GRU4Rec | Caser | SASRec | FMLP-Rec | ConvFormer |
|----------|--------|---------|-------|--------|----------|------------|
| Beauty | H@5 | 0.0164 | 0.0205 | 0.0387 | 0.0398 | **0.0413** |
| | N@5 | 0.0099 | 0.0131 | 0.0249 | 0.0258 | **0.0270** |
| | H@10 | 0.0283 | 0.0347 | 0.0605 | 0.0632 | **0.0675** |
| | N@10 | 0.0137 | 0.0176 | 0.0318 | 0.0333 | **0.0354** |
| | H@20 | 0.0479 | 0.0556 | 0.0902 | 0.0958 | **0.0993** |
| | N@20 | 0.0187 | 0.0229 | 0.0394 | 0.0415 | **0.0433** |
| Sports | H@5 | 0.0129 | 0.0116 | 0.0233 | 0.0218 | **0.0244** |
| | N@5 | 0.0086 | 0.0072 | 0.0154 | 0.0144 | **0.0157** |
| | H@10 | 0.0204 | 0.0194 | 0.0350 | 0.0344 | **0.0387** |
| | N@10 | 0.0110 | 0.0097 | 0.0192 | 0.0185 | **0.0203** |
| | H@20 | 0.0333 | 0.0314 | 0.0507 | 0.0537 | **0.0587** |
| | N@20 | 0.0142 | 0.0126 | 0.0231 | 0.0233 | **0.0253** |
| Toys | H@5 | 0.0097 | 0.0166 | 0.0463 | 0.0456 | **0.0502** |
| | N@5 | 0.0059 | 0.0107 | 0.0306 | 0.0317 | **0.0344** |
| | H@10 | 0.0176 | 0.0270 | 0.0675 | 0.0683 | **0.0753** |
| | N@10 | 0.0084 | 0.0141 | 0.0374 | 0.0391 | **0.0424** |
| | H@20 | 0.0301 | 0.0420 | 0.0941 | 0.0991 | **0.1056** |
| | N@20 | 0.0116 | 0.0179 | 0.0441 | 0.0468 | **0.0500** |
| Yelp | H@5 | 0.0152 | 0.0151 | 0.0162 | 0.0179 | **0.0212** |
| | N@5 | 0.0099 | 0.0096 | 0.0100 | 0.0113 | **0.0137** |
| | H@10 | 0.0263 | 0.0253 | 0.0274 | 0.0304 | **0.0353** |
| | N@10 | 0.0134 | 0.0129 | 0.0136 | 0.0153 | **0.0182** |
| | H@20 | 0.0439 | 0.0422 | 0.0457 | 0.0511 | **0.0566** |
| | N@20 | 0.0178 | 0.0171 | 0.0182 | 0.0205 | **0.0235** |

Table A2: Full-sort performance on the industrial dataset. The best results are marked in bold fonts. Increasing the dropout rate consistently leads to performance drop.

| Methods | Metric | dropout=0.0 | dropout=0.1 | dropout=0.3 | dropout=0.5 |
|---------|--------|-------------|-------------|-------------|-------------|
| SASRec | H@5 | 0.5635 | **0.5662** | 0.5256 | 0.4976 |
| | N@5 | 0.6722 | **0.6747** | 0.6339 | 0.6092 |
| | H@10 | 0.7317 | **0.7335** | 0.6948 | 0.6723 |
| | N@10 | **0.3817** | 0.3751 | 0.3483 | 0.3222 |
| | H@20 | **0.4092** | 0.4026 | 0.3757 | 0.3504 |
| | N@20 | **0.4218** | 0.4152 | 0.3887 | 0.3638 |
| ConvFormer | H@5 | **0.5996** | 0.5836 | 0.5661 | 0.5513 |
| | N@5 | **0.7078** | 0.6937 | 0.6796 | 0.6650 |
| | H@10 | **0.7645** | 0.7519 | 0.7395 | 0.7267 |
| | N@10 | **0.4002** | 0.3879 | 0.3720 | 0.3569 |
| | H@20 | **0.4276** | 0.4158 | 0.4008 | 0.3857 |
| | N@20 | **0.4397** | 0.4282 | 0.4136 | 0.3989 |

- The original implementation of FMLP-Rec may result in a NAN loss function during training and yield subpar performance. To address this issue, we introduce gradient clipping and enhance the initialization process in FMLP-Rec+. These modifications stabilize the training process, and overall performance surpasses that of SASRec given other settings consistent.

- ConvFormer demonstrates the highest overall performance in the CTR estimation task, highlighting the general applicability of the LighTCN layer and the proposed evaluation criteria in various scenarios. Specifically, setting the kernel size as the sequence length (refered to as ConvFormer) without finetuning yields promising results compared to other baseline methods. However, using the full receptive field may not be optimal since early user behaviors could introduce noise and lack informative signals. By fine-tuning the receptive field of the LighTCN layer (referred to as ConvFormer+), further improvements in overall performance can be achieved.

Table A3: Statistics of the employed datasets.

| Dataset | #.Sequences | #.Items | #.Actions | Sparsity |
|---|---|---|---|---|
| Beauty | 22,363 | 12,101 | 198,502 | 99.93% |
| Sports | 25,598 | 18,357 | 296,337 | 99.95% |
| Toys | 19,412 | 11,924 | 167,597 | 99.93% |
| Yelp | 30,431 | 20,033 | 316,354 | 99.95% |
| Industry | 674,491 | 9,690 | 19,699,497 | 99.70% |

Table A4: Performance comparison on our industrial dataset. Bold and underlined fonts indicate the first and second best results, respectively.

| Model | HIT@10 | HIT@20 | HIT@30 | NDCG@10 | NDCG@20 | NDCG@30 |
|---|---|---|---|---|---|---|
| GRU4Rec | 0.5732 | 0.6797 | 0.7365 | 0.3867 | 0.4137 | 0.4258 |
| SASRec | 0.5635 | 0.6722 | 0.7317 | 0.3817 | 0.4092 | 0.4218 |
| FMLP-Rec | 0.5781 | 0.6861 | 0.7449 | 0.3903 | 0.4177 | 0.4302 |
| ConvFormer | **0.5996** | **0.7078** | **0.7645** | **0.4002** | **0.4276** | **0.4397** |

## A.4 Performance of order-sensitive SAR variants

In Table 1 we compare the performance of SAR and its variants, illustrating that the simple yet order-sensitive modules can be competitive alternative to the self-attention token-mixer. We understand that the claims may be aggressive, and it is responsible to ensure the rigor of our experiments. To this end, we have conducted comprehensive experiments on the four benchmarks. We report the results in Table 3, as an extension of Table 1, showing that the superiority of SAR-O, SAR-P and SAR-R holds across a range of critical hyperparameters (the number of blocks) and random seeds (1-10). We will also open-source the code of these variants, along with the training logs and checkpoint models for each seed, to provide empirical support for our claims and facilitate reproducibility.

The main observations from Table 3 are summarized as follows.

- SAR-O, replacing $\mathbf{A}$ in SAR with a trainable parameter matrix $\mathbf{A}^{(\mathrm{O})}$, outperforms SAR on both benchmarks. The superiority is attributed to the order-sensitivity of the parameter matrix.

- SAR-P, personalizing SAR-O's attention matrix to user behavior patterns, achieves similar performance with SAR-O. The incremental improvement suggest that adaptively generated weights in the item-to-item paradigm are not essential for SAR's leading performance.

- SAR-R, fixing SAR-O's weights non-trainable, achieves comparative performance with SAR. Although SAR-R fails to capture semantic relationships between items, it is sensitive to the order of items that is essential to next-item prediction task. As a result, the attention matrix $\mathbf{A}$ in SAR can be replaced with a random matrix $\mathbf{A}^{(\mathrm{R})}$ without performance loss. It is exactly this observation that have inspired our key hypothesis: self-attentive token mixer is not necessarily effective for sequential user modeling. This motivated us to investigate the essences that make Transformer a superior sequential user model, which is one of the major contributions of this work.

- SAR-W is an additional variant that does not appear in the main text, which is similar to SAR-R but has no token-mixer. That is, only the FFN layers are preserved. It exhibits a consistent

Table A5: Performance comparison as the interest extractor on CTR prediction task. Bold and underlined fonts indicate the first and second best results, respectively.

| Interest Extractor | HIT@5 | HIT@10 | HIT@30 | NDCG@5 | NDCG@10 | NDCG@30 |
|---|---|---|---|---|---|---|
| GRU4Rec | 0.4581 | 0.6575 | 0.8887 | 0.2971 | 0.3614 | 0.4173 |
| SASRec | 0.4560 | 0.6713 | 0.8940 | 0.2975 | 0.3677 | 0.4216 |
| FMLP-Rec | 0.3150 | 0.5546 | 0.8494 | 0.2000 | 0.2772 | 0.3479 |
| FMLP-Rec+ | 0.4581 | 0.6649 | 0.8897 | 0.3025 | 0.3689 | 0.4236 |
| ConvFormer | 0.4666 | **0.6787** | **0.8982** | 0.3014 | 0.3703 | 0.4236 |
| ConvFormer+ | **0.5037** | 0.6670 | 0.8929 | **0.3213** | **0.3744** | **0.4290** |

and significant drop in performance compared to SAR. This demonstrates the necessity of token mixing, as even a fixed and randomized token mixer can still maintain competitive performance with SAR.

Notably, the findings above align with recent literature that challenges the role of self-attention in their respective fields. For example, Google's recent work on abstractive summarization found that replacing the attention matrix with a fixed learnable matrix, such as the SAR-R in our work, could improve most metrics over self-attention (Tab. 3 by Tay et al. (2020)). They also concluded that The simplest Synthesizers such as Random Synthesizers are fast competitive baseline. (Section 5.2 by Tay et al. (2020)). Similarly, MLP-Mixer (Tolstikhin et al., 2021) replaces self-attentive token mixers with fixed learnable weights, showing advantages over canonical self-attentive token mixers in many fundamental and data-rich fields.

In fact, our findings build up recent DL advances and make reasonable extensions for sequential user modeling. It has been acknowledged that the token-mixer based on non-trainable fixed matrix could achieve promising performance. For example, Goolge concluded that the non-trainable variant performs achieves a strong 24 BLEU with fixed random attention weights (Tay et al., 2020); replacing self-attention process with a non-trainable fixed Fourier layer (Lee-Thorp et al., 2022) could achieve 80% speed-up with a mere 3%-7% accuracy drop. Furthermore, we note that SAR-R with a fixed random matrix is more lightweight and sensitive to the order of items than SAR, which are quite important for sequential user modeling. Therefore, it is reasonable to observe that the performance gap between SAR and SAR-R is negligible, with SAR-R even outperforming SAR in some cases.

### A.5 PERFORMANCE OF ADDITIONAL ATTENTIVE LIGHT-WEIGHT BASELINES

We have compared ConvFormer with Fastformer (Wu et al., 2021) and PoolingFormer (Zhang et al., 2021) in Table 3, as these two models beat a series of lightweight Transformer variants such as LinFormer (Wang et al., 2020) and LongFormer (Beltagy et al., 2020). However, the results reported in the main text only include two datasets and representative metrics for brevity. We report the results on four datasets and all metrics in Table A7 to make them comprehensive and convenient for reuse, which could facilitate to understand the position of current methods.

## B CONVFORMER-F: ACCELERATION AND APPROXIMATION

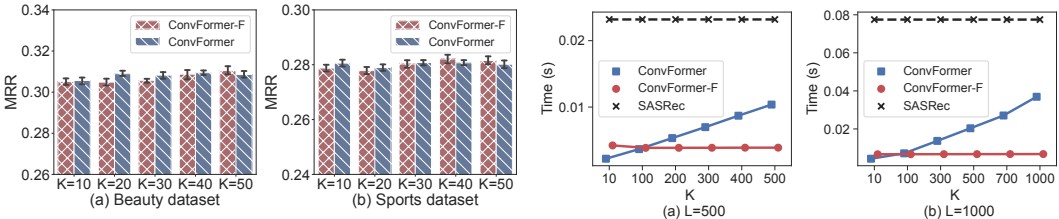

Figure B1: Comparing the recommendation quality (MRR) at different kernel size K.

Figure B2: Comparing the inference time with different sequence length L and kernel size K.

The criterion (1) may lead to inefficiencies in certain scenarios where user behavior sequences are lengthy (Pi et al., 2020; Chen et al., 2021), as the computational complexity approaches $\mathcal{O}(L^2)$ when the receptive field size approaches the sequence length. To handle this complexity, inspired by the convolution theorem (Pratt et al., 2017; Oppenheim et al., 2001) summarized in Lemma 4.1, we develop an acceleration algorithm for ConvFormer, denoted by **ConvFormer-F**. The key idea is that the convolution in the spatial domain can be converted as Hadamard product in the Fourier domain, facilitating more efficient computation of the DWC layer (4) as follows:

$$\text{DWC}(\mathbf{R}) = \mathcal{F}^{-1}\left(\mathcal{F}(\mathbf{R}) \odot \mathcal{F}(\mathbf{C})\right) \tag{10}$$

where $\odot$ indicates the Hadamard point-wise product, $\mathbf{C}$ is right-padded with zeros to ensure the same length with $\mathbf{R}$, $\mathcal{F}$ indicates the Discrete Fourier Transform (DFT), a fundamental technique for processing discrete time series data (Oppenheim et al., 2001), and $\mathcal{F}^{-1}$ indicates the inverse DFT (IDFT). The computational workflow of ConvFormer is summarized in Algorithm 1.

Table A6: Performance comparison of SAR and its variants, as a continued table of Table 1 in the main text. Bold fonts indicate the best performance. Red (resp. green) fonts indicate the variants that are superior (resp. inferior) to SAR with $p$-value $< 0.01$ in paired-sample t-test.

| # Layers | Model | HIT@1 | HIT@5 | HIT@10 | NDCG@5 | NDCG@10 | MRR |
|---|---|---|---|---|---|---|---|
| | | | | **Beauty dataset** | | | |
| | SAR | 0.1771 | 0.3593 | 0.4512 | 0.2726 | 0.3022 | 0.2736 |
| | SAR-P | **0.1778(0.3028)** | 0.3607(0.1581) | 0.4519(0.3028) | 0.2737(0.1920) | 0.3031(0.2002) | 0.2746(0.1975) |
| 1 | SAR-O | 0.1777(0.2720) | **0.3617(0.0554)** | **0.4539(0.2720)** | **0.2743(0.0603)** | **0.3040(0.0165)** | **0.2752(0.0515)** |
| | SAR-R | 0.1772(0.4342) | 0.3602(0.2869) | 0.4521(0.4342) | 0.2733(0.2821) | 0.3029(0.2267) | 0.2743(0.2496) |
| | SAR-W | 0.1703(0.9999) | 0.3452(1.0000) | 0.4338(0.9999) | 0.2622(1.0000) | 0.2907(1.0000) | 0.2641(1.0000) |
| | SAR | 0.1816 | 0.3690 | 0.4631 | 0.2800 | 0.3104 | 0.2804 |
| | SAR-P | **0.1820(0.3520)** | **0.3696(0.3163)** | 0.4635(0.3520) | **0.2805(0.2861)** | **0.3108(0.3262)** | **0.2809(0.2848)** |
| 2 | SAR-O | 0.1804(0.8409) | 0.3688(0.5476) | **0.4636(0.8409)** | 0.2791(0.7734) | 0.3097(0.7387) | 0.2795(0.7907) |
| | SAR-R | 0.1818(0.4456) | 0.3687(0.6400) | 0.4614(0.4456) | 0.2797(0.6307) | 0.3096(0.7919) | 0.2801(0.6159) |
| | SAR-W | 0.1697(1.0000) | 0.3450(1.0000) | 0.4342(1.0000) | 0.2616(1.0000) | 0.2903(1.0000) | 0.2635(1.0000) |
| | SAR | 0.1822 | 0.3709 | 0.4656 | 0.2810 | 0.3116 | 0.2814 |
| | SAR-P | 0.1830(0.2599) | 0.3740(0.0308) | 0.4684(0.2599) | 0.2831(0.0507) | 0.3136(0.0598) | 0.2830(0.0757) |
| 3 | SAR-O | **0.1841(0.0713)** | **0.3748(0.0048)** | **0.4695(0.0713)** | **0.2841(0.0084)** | **0.3147(0.0061)** | **0.2841(0.0127)** |
| | SAR-R | 0.1825(0.4377) | 0.3724(0.1134) | 0.4666(0.4377) | 0.2820(0.2157) | 0.3125(0.2469) | 0.2822(0.2712) |
| | SAR-W | 0.1726(1.0000) | 0.3477(1.0000) | 0.4379(1.0000) | 0.2644(1.0000) | 0.2935(1.0000) | 0.2665(1.0000) |
| | | | | **Sports dataset** | | | |
| | SAR | 0.1391 | 0.3316 | 0.4490 | 0.2382 | 0.2761 | 0.2424 |
| | SAR-P | 0.1403(0.1274) | 0.3339(0.0702) | 0.4526(0.1274) | 0.2399(0.0881) | 0.2782(0.0528) | 0.2440(0.0700) |
| 1 | SAR-O | **0.1416(0.0102)** | **0.3344(0.0246)** | **0.4528(0.0102)** | **0.2409(0.0081)** | **0.2790(0.0045)** | **0.2450(0.0054)** |
| | SAR-R | 0.1410(0.0457) | 0.3333(0.1171) | 0.4506(0.0457) | 0.2399(0.0541) | 0.2777(0.0481) | 0.2440(0.0403) |
| | SAR-W | 0.1328(0.9997) | 0.3148(1.0000) | 0.4275(0.9997) | 0.2264(1.0000) | 0.2628(1.0000) | 0.2318(1.0000) |
| | SAR | 0.1443 | 0.3442 | 0.4647 | 0.2473 | 0.2861 | 0.2504 |
| | SAR-P | **0.1464(0.0051)** | **0.3480(0.0003)** | **0.4686(0.0051)** | **0.2503(0.0002)** | **0.2891(0.0001)** | **0.2531(0.0003)** |
| 2 | SAR-O | 0.1462(0.0300) | 0.3474(0.0021) | 0.4682(0.0300) | 0.2497(0.0059) | 0.2887(0.0062) | 0.2526(0.0100) |
| | SAR-R | 0.1446(0.3857) | 0.3438(0.7097) | 0.4646(0.3857) | 0.2470(0.6650) | 0.2860(0.5882) | 0.2504(0.5743) |
| | SAR-W | 0.1335(1.0000) | 0.3175(1.0000) | 0.4328(1.0000) | 0.2282(1.0000) | 0.2653(1.0000) | 0.2335(1.0000) |
| | SAR | 0.1451 | 0.3485 | 0.4709 | 0.2498 | 0.2893 | 0.2528 |
| | SAR-P | 0.1492(0.0042) | 0.3550(0.0009) | 0.4792(0.0042) | 0.2552(0.0021) | 0.2952(0.0007) | 0.2576(0.0021) |
| 3 | SAR-O | **0.1494(0.0127)** | **0.3560(0.0008)** | **0.4794(0.0127)** | **0.2558(0.0020)** | **0.2956(0.0005)** | **0.2581(0.0025)** |
| | SAR-R | 0.1474(0.0750) | 0.3531(0.0041) | 0.4766(0.0750) | 0.2533(0.0173) | 0.2931(0.0047) | 0.2559(0.0189) |
| | SAR-W | 0.1339(1.0000) | 0.3177(1.0000) | 0.4326(1.0000) | 0.2285(1.0000) | 0.2655(1.0000) | 0.2338(1.0000) |
| | | | | **Toy dataset** | | | |
| | SAR | 0.1738 | 0.3517 | 0.4475 | 0.2666 | 0.2975 | 0.2690 |
| | SAR-P | 0.1760(0.0687) | 0.3565(0.0004) | **0.4520(0.0687)** | 0.2701(0.0030) | 0.3009(0.0048) | 0.2720(0.0098) |
| 1 | SAR-O | **0.1765(0.0037)** | **0.3567(0.0000)** | 0.4515(0.0037) | **0.2705(0.0000)** | **0.3011(0.0000)** | **0.2723(0.0001)** |
| | SAR-R | 0.1771(0.0040) | 0.3542(0.0020) | 0.4486(0.0040) | 0.2695(0.0006) | 0.3000(0.0013) | 0.2719(0.0007) |
| | SAR-W | 0.1712(0.9606) | 0.3411(1.0000) | 0.4324(0.9606) | 0.2598(0.9999) | 0.2893(1.0000) | 0.2630(0.9998) |
| | SAR | 0.1790 | 0.3647 | 0.4616 | 0.2759 | 0.3071 | 0.2771 |
| | SAR-P | **0.1816(0.0383)** | **0.3656(0.2032)** | 0.4620(0.0383) | **0.2776(0.0714)** | **0.3087(0.0529)** | **0.2790(0.0515)** |
| 2 | SAR-O | 0.1801(0.1893) | 0.3650(0.3160) | **0.4625(0.1893)** | 0.2766(0.1591) | 0.3080(0.1069) | 0.2780(0.1196) |
| | SAR-R | 0.1797(0.2902) | 0.3634(0.9093) | 0.4602(0.2902) | 0.2755(0.6473) | 0.3068(0.6494) | 0.2770(0.5346) |
| | SAR-W | 0.1697(0.9998) | 0.3410(1.0000) | 0.4345(0.9998) | 0.2590(1.0000) | 0.2891(1.0000) | 0.2622(1.0000) |
| | SAR | 0.1811 | 0.3681 | 0.4661 | 0.2786 | 0.3102 | 0.2796 |
| | SAR-P | **0.1830(0.0929)** | **0.3703(0.018)** | **0.4691(0.0929)** | **0.2807(0.0103)** | **0.3126(0.0059)** | **0.2818(0.0105)** |
| 3 | SAR-O | 0.1824(0.2662) | 0.3698(0.0980) | 0.4673(0.2662) | 0.2801(0.1055) | 0.3115(0.1071) | 0.2810(0.1538) |
| | SAR-R | 0.1792(0.8547) | 0.3664(0.9009) | 0.4643(0.8547) | 0.2769(0.9272) | 0.3085(0.9502) | 0.2780(0.9044) |
| | SAR-W | 0.1690(1.0000) | 0.3412(1.0000) | 0.4345(1.0000) | 0.2587(1.0000) | 0.2888(1.0000) | 0.2617(1.0000) |
| | | | | **Yelp dataset** | | | |
| | SAR | 0.2199 | 0.5552 | 0.7315 | 0.3922 | 0.4494 | 0.3761 |
| | SAR-P | 0.2208(0.2471) | 0.5583(0.0285) | **0.7332(0.2471)** | 0.3944(0.0520) | 0.4511(0.0363) | 0.3777(0.0723) |
| 1 | SAR-O | **0.2216(0.0698)** | **0.5589(0.0095)** | 0.7331(0.0698) | **0.3950(0.0221)** | **0.4515(0.0230)** | **0.3783(0.0262)** |
| | SAR-R | 0.2199(0.5141) | 0.5570(0.1078) | 0.7313(0.5141) | 0.3931(0.2295) | 0.4495(0.4106) | 0.3764(0.3732) |
| | SAR-W | 0.2026(1.0000) | 0.5226(1.0000) | 0.6969(1.0000) | 0.3665(1.0000) | 0.4230(1.0000) | 0.3537(1.0000) |
| | SAR | 0.2254 | 0.5684 | 0.7446 | 0.4018 | 0.4589 | 0.3842 |
| | SAR-P | 0.2290(0.0003) | 0.5731(0.0000) | 0.7473(0.0003) | 0.4061(0.0000) | 0.4626(0.0001) | 0.3878(0.0001) |
| 2 | SAR-O | 0.2281(0.0267) | 0.5713(0.0459) | 0.7472(0.0267) | 0.4048(0.0220) | 0.4618(0.0094) | 0.3870(0.0010) |
| | SAR-R | 0.2275(0.0895) | 0.5692(0.2404) | 0.7455(0.0895) | 0.4033(0.1147) | 0.4604(0.0861) | 0.3858(0.0795) |
| | SAR-W | 0.2070(1.0000) | 0.5271(1.0000) | 0.7013(1.0000) | 0.3708(1.0000) | 0.4272(1.0000) | 0.3577(1.0000) |
| | SAR | 0.2291 | 0.5751 | 0.7498 | 0.4071 | 0.4638 | 0.3886 |
| | SAR-P | 0.2308(0.1261) | 0.5787(0.0376) | **0.7539(0.1261)** | 0.4098(0.0720) | 0.4666(0.0366) | 0.3908(0.0746) |
| 3 | SAR-O | 0.2324(0.0048) | 0.5800(0.0038) | 0.7538(0.0048) | 0.4115(0.0019) | 0.4679(0.0004) | 0.3924(0.0011) |
| | SAR-R | 0.2314(0.0186) | 0.5784(0.0332) | 0.7525(0.0186) | 0.4098(0.0255) | 0.4663(0.0118) | 0.3908(0.0167) |
| | SAR-W | 0.2045(1.0000) | 0.5257(1.0000) | 0.7014(1.0000) | 0.3693(1.0000) | 0.4262(1.0000) | 0.3564(1.0000) |

Table A7: Comparison with emerging attentive light-weight methods. The bold fonts represent the best performance. "*" marks the metrics that ConvFormer improves significantly over the best baselines, with p-value $< 0.01$ in the paired sample t-test.

| Dataset | Model | H@1 | H@5 | H@10 | N@5 | N@10 | MRR |
|---------|-------|-----|-----|------|-----|------|-----|
| Beauty | SASRec | 0.1870 | 0.3741 | 0.4696 | 0.2848 | 0.3156 | 0.2852 |
| | FastFormer | 0.1405 | 0.3395 | 0.4454 | 0.2438 | 0.2780 | 0.2449 |
| | PoolingFormer | 0.1930 | 0.3932 | 0.4925 | 0.2981 | 0.3302 | 0.2971 |
| | ConvFormer | **0.2019*** | **0.4119*** | **0.5105*** | **0.3125*** | **0.3443*** | **0.3093*** |
| Sports | SASRec | 0.1445 | 0.3466 | 0.4622 | 0.2497 | 0.2869 | 0.2520 |
| | FastFormer | 0.1185 | 0.3249 | 0.4573 | 0.2238 | 0.2665 | 0.2284 |
| | PoolingFormer | 0.1568 | 0.3741 | 0.5000 | 0.2687 | 0.3093 | 0.2693 |
| | ConvFormer | **0.1671*** | **0.3891*** | **0.5116*** | **0.2819*** | **0.3215*** | **0.2808*** |
| Toys | SASRec | 0.1878 | 0.3682 | 0.4663 | 0.2820 | 0.3136 | 0.2842 |
| | FastFormer | 0.1301 | 0.3390 | 0.4517 | 0.2380 | 0.2744 | 0.2384 |
| | PoolingFormer | 0.1893 | 0.3873 | 0.4893 | 0.2927 | 0.3256 | 0.2925 |
| | ConvFormer | **0.2007*** | **0.4033*** | **0.5100*** | **0.3069*** | **0.3384*** | **0.3048*** |
| Yelp | SASRec | 0.2375 | 0.5745 | 0.7373 | 0.4113 | 0.4642 | 0.3927 |
| | FastFormer | 0.1918 | 0.5451 | 0.7355 | 0.3727 | 0.4344 | 0.3557 |
| | PoolingFormer | 0.2539 | 0.6087 | 0.7663 | 0.4378 | 0.4890 | 0.4144 |
| | ConvFormer | **0.2816*** | **0.6347*** | **0.7863*** | **0.4653*** | **0.5146*** | **0.4406*** |

**Definition B.1** (DFT and IDFT). *Given an* L*-length sequence* $\mathbf{X} = [x_1, ..., x_L]$*, DFT projects it to a set of predefined exponential basis, and the projection onto the* $k$*-th basis is calculated as*

$$x_k^{(F)} = \mathcal{F}(\mathbf{X})_k = \sum_{l=0}^{L-1} x_l \exp(-\frac{2\pi i}{L}lk), \quad 0 \leq k \leq L-1, \tag{11}$$

*where* $\exp(\cdot)$ *is the exponential basis,* $i$ *is the imaginary unit,* $k$ *indicates the frequency of the exponential basis. Inversely, given the projection onto each basis, we can recover the original sequence via the Inverse DFT (IDFT):*

$$x_l = \mathcal{F}^{-1}(\mathbf{X}^{(F)})_l = \frac{1}{L} \sum_{k=0}^{L-1} x_k^{(F)} \exp(\frac{2\pi i}{L}lk), \quad 0 \leq l \leq L-1, \tag{12}$$

**Lemma B.1.** *Let* $\mathbf{X} = [x_1, \ldots, x_L]$ *and* $\mathbf{C} = [c_1, \ldots, c_L]$ *be two* L*-length sequences. The Fourier transform of a convolution of the two signals is the Hadamard product of their Fourier transforms:*

$$\mathcal{F}(\mathbf{C} * \mathbf{X}) = \mathcal{F}(\mathbf{C}) \odot \mathcal{F}(\mathbf{X}). \tag{13}$$

*Proof.* Assuming that the sequence $\mathbf{X}$ is periodic with the period L, we have:

$$
\begin{aligned}
\mathcal{F}(\mathbf{C} * \mathbf{X})_k &\overset{(a)}{=} \sum_{l=0}^{L-1} \left( \left( \sum_{j=0}^{L-1} c_j x_{l-j} \right) \exp\left( -\frac{2\pi i}{L}lk \right) \right) \\
&= \sum_{j=0}^{L-1} c_j \left( \sum_{l=0}^{L-1} x_{l-j} \exp\left( -\frac{2\pi i}{L}lk \right) \right) \\
&\overset{(b)}{=} \sum_{j=0}^{L-1} c_j \exp\left( -\frac{2\pi i}{L}jk \right) \left( \sum_{l=0}^{L-1} x_{l-j} \exp\left( -\frac{2\pi i}{L}(l-j)k \right) \right) \\
&\overset{(c)}{=} \left( \sum_{j=0}^{L-1} c_j \exp\left( -\frac{2\pi i}{L}jk \right) \right) \left( \sum_{l=0}^{L-1} x_l \exp\left( -\frac{2\pi i}{L}lk \right) \right) \\
&\overset{(d)}{=} \mathcal{F}(\mathbf{C})_k \odot \mathcal{F}(\mathbf{X})_k.
\end{aligned} \tag{14}
$$

Below is some explanation for the derivation:

---

**Algorithm 1** The computational workflow of ConvFormer

---

**Input**: a user's sequence $S = \{i_1, \cdots, i_L\}$, a target item $i_t$.
**Output**: the preference score $p(i_t|i_{1:L})$.

1:  get input embeddings $\hat{\mathbf{E}}$ of $S$ by Eq.(3)
2:  set $\mathbf{R} \leftarrow \hat{\mathbf{E}}$ and lookup target item's embedding $\mathbf{e}_t$
3:  **for** $n = 1$ to N **do**                                    ▷ Stacking N LighTCN layers
4:      **if** acceleration **then**                              ▷ Based on Eq.(9)
5:          $\mathbf{C}^{(F)} \leftarrow \mathcal{F}(\text{Pad}(\mathbf{C})), \quad \mathbf{R}^{(F)} \leftarrow \mathcal{F}(\mathbf{R})$
6:          $\hat{\mathbf{R}} \leftarrow \text{LayerNorm}(\mathbf{R} + \text{Dropout}(\mathcal{F}^{-1}(\mathbf{C}^{(F)} \odot \mathbf{R}^{(F)})))$
7:      **else**                                                 ▷ Standard operation with Eq.(4)
8:          $\hat{\mathbf{R}} \leftarrow \text{LayerNorm}(\mathbf{R} + \text{Dropout}(\text{DWC}(\mathbf{R})))$
9:      $\tilde{\mathbf{R}} \leftarrow \text{LayerNorm}(\hat{\mathbf{R}} + \text{Dropout}(\text{CWC}(\hat{\mathbf{R}})))$
10:     $\mathbf{R} \leftarrow \tilde{\mathbf{R}}$
11: $p(i_t|i_{1:L}) = \mathbf{e}_t^\top \mathbf{R}[L]$                    ▷ Dot product scorer

---

(a) is the definition of DFT and discrete convolution operation;

(b) breaks down $\exp(2\pi ilk/L)$ into $\exp(2\pi i(l-j)k/L)$ and $\exp(2\pi ijk/L)$;

(c) holds due to the periority of $\mathbf{X}$ and $\exp(\cdot)$;

(d) is the definition of DFT.

Eq.(14) holds for all $0 \leq k \leq L - 1$. The proof is completed.                    □

Both DFT and IDFT can be implemented as matrix-vector multiplication, which is differentiable and thus can be integrated in neural user models. However, the complexity of DFT and IDFT is $\mathcal{O}(L^2)$, with no theoretical superiority over the standard DWC layer. In this regard, the actual accelerator are the Fast Fourier Transform (FFT) and its inverse, which calculate DFT and IDFT in a recursive manner and reduce their complexity to $\mathcal{O}(L \log(L))$. In this way, we can reduce the computational complexity of the DWC layer from $\mathcal{O}(L^2)$ to $\mathcal{O}(L \log(L))$, which is extremely advantageous when modelling long user behavior sequences.

To showcase the efficacy of ConvFormer-F, it is necessary to verify its accuracy equivalence and speed acceleration with respect to the standard ConvFormer. Overall, we reuse the hyperparameters in Table C8, but for the stability of test results, we set the batch size to 512. To emphasize the difference in speed, we omit the inference time of these methods' common layers, including embedding and FFN layers. In practice, we firstly generate a random matrix $\mathbf{R} \in \mathbb{R}^{L \times D}$ and then feed it into a self-attention layer, a CWC layer and its accelerated version. We vary the maximum sequence length L in $\{500, 1000\}$ and the convolution kernel size from 10 to L, and record the average GPU inference time over 10 runs. Experiments are conducted with an AMD EPYC 7742 64-Core processor and an NVIDIA RTX A6000 GPU.

The accuracy comparison is conducted in Figure B1. Overall, there is no significant difference between the two methods, as evidenced by the substantial overlap between the 95% confidence intervals represented by the error bars. Precisely, the MRR differences at $K = 40$ are merely $8e^{-4}$ and $1e^{-3}$ for the beauty and sports datasets, respectively. These observations support the accuracy equivalence between ConvFormer and ConvFormer-F.

The GPU inference time is compared in Figure B2. The y-axis indicates the total inference time for a batch of 512 sequences. Overall, the inference time of ConvFormer rises linearly with respect to the kernel size K, while that of ConvFormer-F keeps constant with respect to K. As a result, the speedup of the fast approximation approach is not readily apparent for small kernel sizes, *e.g.,* $K < 100$ in $L = 1000$; nonetheless, as the kernel size is increased for better accuracy, the superiority of ConvFormer-F becomes more pronounced. Note that the inference cost is a major flaw with SASRec, which is mostly brought on by its item-to-item paradigm and softmax operator. In particular, ConvFormer and ConvFormer-F accelerate SASRec by 3x and 5x, respectively, even with the largest kernel setting, *i.e.,* $K = L$.

## C    REPRODUCTION DETAILS

### C.1    BASELINE DESCRIPTION

In our evaluation, ConvFormer is benchmarked against a range of established baselines in the field. We adhered to the experimental settings outlined by Zhou et al. (2022), including hyperparameters and training protocols, to ensure consistency and comparability with the established benchmark (refer to Table C8 for detailed hyperparameter settings). The baseline models include:

- **PopRec** is a ranking model based on item popularity, determined by the frequency of interactions;
- **FM** (Rendle, 2010) is a factorized model to characterize pairwise interactions between variables;
- **AutoInt** (Song et al., 2019) employs self-attention for automatic feature interaction;
- **GRU4Rec** (Hidasi & Karatzoglou, 2018) encodes user interests with stacked gated recurrent unit;
- **Caser** (Tang & Wang, 2018) encodes user interests with horizontal and vertical convolution layers;
- **HGN** (Huang et al., 2020) uses hierarchical gating to model personalized long-short term interests;
- **RepeatNet** (Ren et al., 2019) strengthens RNN with a repetition mechanism for adaptive item selection from user behaviors;
- **CLEA** (Qin et al., 2021) involves an item-level denoising procedure through contrastive learning;
- **SASRec** (Kang & McAuley, 2018) utilizes self-attentive token mixer to capture behavior patterns;
- **BERT4Rec** (Sun et al., 2019) extends SASRec with bidirectional encoders and Cloze training;
- **SRGNN** (Wu et al., 2019) session-based GNN to characterize item transitions for prediction;
- **GCSAN** (Xu et al., 2019) adds GNN to SASRec to encode dependencies between nearby items;
- **FMLP-Rec** (Zhou et al., 2022) replaces the self-attentive token mixer in SASRec with a learnable filter layer, which is the state-of-the-art approach in the context of sequential user modeling.

Additionally, in Section 5.3, we compare ConvFormer with advanced attentive token mixers: Fastformer (Wu et al., 2021) and PoolingFormer (Zhang et al., 2021), which have demonstrated effectiveness over several Transformer variants (Wang et al., 2020; Beltagy et al., 2020).

- **PoolingFormer** implements a localized and large receptive self-attention layer, using pooling to accelerate key and value vector computations.
- **FastFormer** utilizes an additive attention mechanism to model global context, and transforms each item's representation based on its interaction with global context representations.

We replicated Fastformer using its open-source implementation at `https://github.com/wuch15/Fastformer`and developed PoolingFormer from scratch. For a fair comparison, common hyperparameters between ConvFormer, SASRec, and these models were kept consistent, such as learning rate and hidden dimensions. Individual hyperparameters, like the pooling size in PoolingFormer, were finely tuned for optimal performance. All of these experiments are repeated 11 times with different random seeds[4].

### C.2    DESCRIPTION OF SAR AND ITS VARIANTS

In Section 3, we commence our analysis with the standard SAR model, which serves as the basis for our experimental variants. The SAR model operates on an input representation sequence $\mathbf{R} \in \mathbb{R}^{L \times D}$ and employs an attention matrix $\mathbf{A}$ to fuse contextual information within the value vector: $\mathbf{S} = \mathbf{A}(\mathbf{R}\mathbf{W}^{(V)})$. This is followed by a feed-forward network (FFN) for cross-channel intersection. $\mathbf{A}$ is computed through an item-to-item paradigm as follows:

$$\mathbf{Q} = \mathbf{R}\mathbf{W}^{(Q)} + \mathbf{b}^{(Q)},$$
$$\mathbf{K} = \mathbf{R}\mathbf{W}^{(K)} + \mathbf{b}^{(K)},$$
$$\mathbf{A} = \mathrm{softmax}(\mathbf{Q}\mathbf{K}^{\top}\sqrt{D}).$$

---

[4] Results are reported with seeds 1-10 and 42.

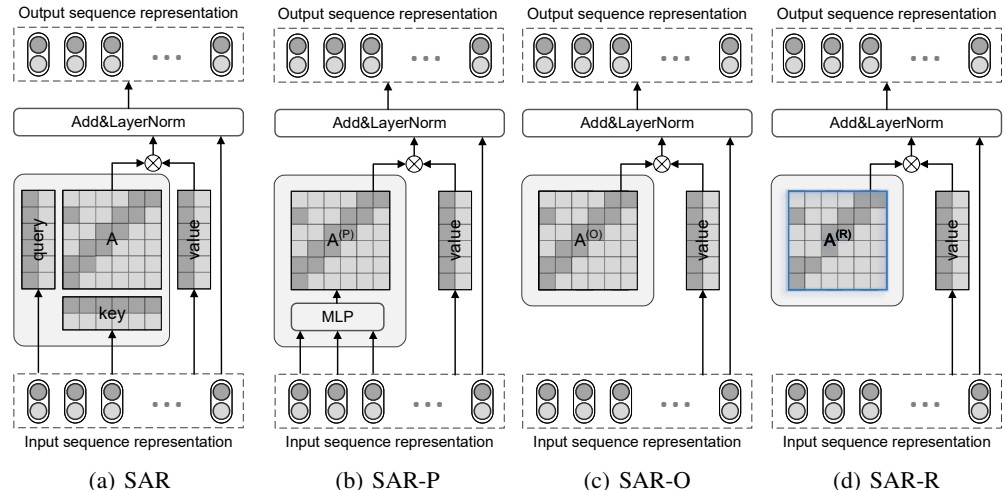

Figure C3: Simple yet order-sensitive architectures for the alternatives to the item-to-item paradigm in SAR. The non-trainable parameters are indicated by the blue box in (d).

To perform a valid comparison with its alternative counterparts, the multi-head trick is disabled in this implementation *i.e.,* the number of heads is set to 1 as indicated in Table C8. Other components, such as skip connections, dropout, and layer normalization, are retained in line with the canonical SASRec model (Kang & McAuley, 2018).

### C.2.1 ORDER-SENSITIVE VARIANTS.

In this section, we describe order-sensitive SAR variants used in Section 3.1. These variants, as graphically illustrated in Figure C3, maintain the standard SAR framework but differ in how the attention matrix $\mathbf{A}$ is computed.

- **SAR-O** employs a trainable parameter matrix $\mathbf{A}^{(O)} \in \mathbb{R}^{L \times L}$ as the attention matrix.
- **SAR-P** utilizes a trainable MLP module to dynamically generate the attention scores, providing a more customized output. The attention scores generated by each item is $\mathbf{A}_l^{(P)} = \mathbf{R}_l \mathbf{W}^{(P)} + \mathbf{b}^{(P)}, \quad 0 \leq l \leq L$, where $\mathbf{W}^{(P)} \in \mathbb{R}^{D \times T}$ and $\mathbf{b}^{(P)} \in \mathbb{R}^T$ are shared parameters for all items.
- **SAR-R** is similar to SAR-O, but attention $\mathbf{A}^{(R)}$ is randomly initialized, fixed, and non-trainable.

**Justification of order sensitivity.** The concept of order sensitivity in a token mixer can be defined as follows: for any item in the input sequence, changing the order of other items alters the representation outputted by the token mixer. To illustrate this, consider the representation of the $i$-th item in the input sequence, denoted as $\mathbf{r}_i \in \mathbb{R}^D$. In SAR and its variants, the output representation of $\mathbf{r}_i$ is given by $\mathbf{s}_i = \sum a_{i,j} \mathbf{r}_j$ [5]. If we switch the order of two items, $j_1$ and $j_2$, the output representation changes accordingly: $\mathbf{s}'_i = \sum_{j=1, j \neq j_1, j \neq j_2}^{L} a_{i,j} \mathbf{r}_j + a_{i,j_1} \mathbf{r}_{j_2} + a_{i,j_2} \mathbf{r}_{j_1}$, reflecting the model's sensitivity to the item order.

- In SAR, $\mathbf{s}_i$ does not exhibit order sensitivity. This is because the attention weights are dynamically determined based on the similarity between items, irrespective of their positions in the sequence. Consequently, swapping two items, $j_1$ and $j_2$, does not alter the output representation of the $i$-th item, i.e., $\mathbf{s}'_i = \mathbf{s}_i$.
- In SAR-O and SAR-R, the weights $a_{i,j}$ are fixed with respect to positions rather than dynamically calculated, which makes $\mathbf{s}_i$ sensitive to the order of items, manifested as $\mathbf{s}'_i \neq \mathbf{s}_i$.

---

[5] We omit the value mapping for brevity here. Since the value mapping is performed on each time step of $\mathbf{r}_j$ individually, independent with the order of input sequence, ignoring it does not affect order sensitivity analysis.

- In SAR-P, the only difference from SAR-O is that the weights $\mathbf{a}_i$ are dependent on $\mathbf{r}_i$: $\mathbf{a}_i = [a_{i,j}]_{j=1:\text{L}} = \text{MLP}(\mathbf{r}_i)$. Since the generated weights remain dependent on the position index $j$, $\mathbf{s}'_i \neq \mathbf{s}_i$ holds and thus SAR-P is order-sensitive.

### C.2.2 Variants with non-shared parameters.

To explore the effect of parameter sharing and the resulting overall lightweight architecture, we introduce two additional SAR variants:

- **SAR-N** is similar to SAR, but the query, key and value mappings are distinct at different steps:

$$\mathbf{Q}_l^{(\text{N})} = \mathbf{RW}^{(\text{Q})}[l] + \mathbf{b}^{(\text{Q})}[l], \quad 0 \leq l \leq \text{L},$$
$$\mathbf{K}_l^{(\text{N})} = \mathbf{RW}^{(\text{K})}[l] + \mathbf{b}^{(\text{K})}[l], \quad 0 \leq l \leq \text{L},$$
$$\mathbf{A}^{(\text{N})} = \text{softmax}(\mathbf{Q}^{(\text{N})}\mathbf{K}^{(\text{N})\top}/\sqrt{\text{D}}).$$

- **SAR-N+** utilizes all input items to generate query, key, and value vectors. Precisely, let $\mathcal{T} : \mathbb{R}^{\text{T}\times\text{D}} \to \mathbb{R}^{\text{TD}\times 1}$ be the flatten operator and $\mathcal{T}^{-1}$ be its inverse, the attention is generated by:

$$\mathbf{Q}^{(\text{N}+)} = \mathcal{T}^{-1}\left(\mathcal{T}(\mathbf{R})\mathbf{W}^{(\text{Q}+)} + \mathbf{b}^{(\text{Q}+)}\right),$$
$$\mathbf{K}^{(\text{N}+)} = \mathcal{T}^{-1}\left(\mathcal{T}(\mathbf{R})\mathbf{W}^{(\text{K}+)} + \mathbf{b}^{(\text{K}+)}\right),$$
$$\mathbf{A}^{(\text{N}+)} = \text{softmax}(\mathbf{Q}^{(\text{N}+)}\mathbf{K}^{(\text{N}+)\top}/\sqrt{\text{D}}).$$

where $\mathbf{W}^{(\text{Q}+)} \in \mathbb{R}^{\text{TD}\times\text{TD}}, \mathbf{W}^{(\text{K}+)} \in \mathbb{R}^{\text{TD}\times\text{TD}}, \mathbf{b}^{(\text{Q}+)} \in \mathbb{R}^{\text{TD}\times 1}, \mathbf{b}^{(\text{K}+)} \in \mathbb{R}^{\text{TD}\times 1}$.

### C.3 Hyperparameter Setting

Table C8 presents the configuration of parameters. The report of model performance and the fine-tuning of hyperparameters follow the protocol as follows.

- For baselines with publicly available results on the benchmark, we use the results reported based on the work by Zhou et al. (2022). We have checked the results reproduciable and adhered to the same experimental setting.

- For baselines without available results, we reproduce them with careful tuning. For fairness, we maintain the same values for the common hyperparameters with current SOTA baseline and ConvFormer, e.g., the embedding dimension, learning rate, dropout ratio. Then, we finetune their own hyperparameters, e.g., the pooling size in PoolingFormer.

- For SAR-variants in Section A.4, these SAR variants rarely introduce new hyperparameters. To make a fair comparison, we use the same common hyperparameters with SAR.

- For Convformer, we use the same values of common hyperparameters such as learning rate, number of blocks and dropout rate with the current SOTA baseline by Zhou et al. (2022). Fine-tuning of ConvFormer is limited to its own hyperparameter, K, as detailed in Table C8. Exhaustive tuning of other hyperparams is disabled.

## D Connecting the dots

### D.1 Attention v.s. Convolution, a moving average perspective

The efficacy of model simplification often hinges on precise prior knowledge, prompting an inquiry into why certain simplifications to the Transformer architecture prove effective and what insights they offer. The self-attentive architecture can be conceptualized as a moving average (MA) model in the value sequence $\mathbf{V}$ as $\mathbf{S} = \mathbf{AV}$, where the weights $\mathbf{A}$ are dynamically generated from the inputs. While the dynamic weight augments model capacity, it encounters limitations in sequential user modeling: (1) order-sensitivity; (2) instability, where the model parameters vary with the input, leading to difficulties in parameter identification. SAR-O, which introduces an input-independent

Table C8: Parameter configurations on each dataset. "*" in the Model field is a wildcard for models.

| Parameter | Model | Beauty | Sports | Toys | Yelp |
|---|---|---|---|---|---|
| max sequence length, L | * | 50 | 50 | 50 | 50 |
| number of layers, N | * | 2 | 2 | 2 | 2 |
| hidden dimension, D | * | 64 | 64 | 64 | 64 |
| convolution kernel size, K | ConvFormer | 45 | 30 | 30 | 30 |
| padding mode | ConvFormer | circular | circular | circular | circular |
| learning rate | * | $1e^{-3}$ | $1e^{-3}$ | $1e^{-3}$ | $1e^{-3}$ |
| weight decay | * | 0 | 0 | 0 | 0 |
| number of attention heads | SAR | 1 | 1 | 1 | 1 |
| number of attention heads | PoolingFormer | 2 | 2 | 2 | 2 |
| number of attention heads | FastFormer | 4 | 2 | 2 | 2 |
| attention dropout probability | * | 0.5 | 0.5 | 0.5 | 0.5 |
| hidden dropout probability | * | 0.5 | 0.5 | 0.5 | 0.5 |
| batch size | * | 256 | 256 | 256 | 256 |
| patience | * | 10 | 10 | 10 | 10 |
| number of maximum epochs | * | 200 | 200 | 200 | 200 |
| pooling size | PoolingFormer | 2 | 4 | 2 | 4 |
| pooling stride | PoolingFormer | 2 | 4 | 2 | 4 |
| local convolution kernel size | PoolingFormer | 10 | 10 | 20 | 20 |

parameter matrix, effectively mitigates these limitations and enhances performance. This suggests that traditional MA models retain significant research and practical value in this domain.

The MA model is designed for Markov process identification problems. Markov processes have two important properties: (1) order, i.e. the number of the previous steps related to the current state, and (2) coupling, i.e. whether the update of a particular channel depends on the states in other channels. Our findings indicate that increasing the receptive field and minimizing channel coupling improve performance, suggesting that user behavior sequences in latent space exhibit high-order, decoupled Markovian properties. Techniques like optimal filtering and smoothing offer robust inference methods for such processes, as exemplified by the success of Kalman attention by Liu et al. (2020), which opens up promising avenues for future research.

### D.2 LARGE RECEPTIVE FIELD V.S. LIGHTWEIGHT ARCHITECTURE, A LEARNING THEORY PERSPECTIVE

In Criteria 2 and 3, we advocate for a large receptive field coupled with a lightweight architecture as essential elements for effective token mixers. While our experimental results substantiate these criteria, we further elucidate their theoretical underpinnings through the lens of Probably Approximately Correct (PAC) learning, based on Lemma D.1.

**Lemma D.1.** *Let $g$ be the hypothesis (model) selected by the learning algorithm with the 'statistical' large dataset $\mathcal{D}$. Let $E_{\text{in}}(g)$ and $E_{\text{out}}(g)$ be the within-sample error and out-of-sample error of the selected model $g$, the generalization gap is defined as*

$$\delta(g) := E_{\text{out}}(g) - E_{\text{in}}(g). \tag{15}$$

*If $\delta$ is significantly large, overfitting happens and learning fails. Furthermore, $g$ can be bounded as:*

$$P_{\mathcal{D}}[\underbrace{|E_{\text{out}}(g) - E_{\text{in}}(g)|}_{\delta(g)} > \epsilon] \leq 4(2M) \exp\left(-\frac{1}{8}\epsilon^2 M\right), \tag{16}$$

*where $M$ is the sample size of the dataset $\mathcal{D}$, $d_{\text{vc}}$ is the VC-dimension that measures the complexity of the model, $\epsilon$ is a confidence threshold.*

Employing a large receptive field enables efficient capture of long-term patterns, offering advantage over stacking multiple small kernels which can distort information through successive non-linear transformations, thus enabling a reduced within-sample error $E_{\text{in}}$. However, this approach increases model complexity measured by $d_{\text{vc}}$, thereby widening the generalization gap $\delta$. To mitigate this, a lightweight architecture, achieved through techniques like parameter sharing or inductive bias, is

Table E9: DWC vs convolution-based sequential user models (Tang & Wang, 2018).

| | DWC layer | Horizontal convolution | Vertical convolution | Advantage of the DWC layer |
|---|---|---|---|---|
| # convolution kernel | 1 | Z | Z | The DWC layer is more lightweight |
| Convolution kernel | depth-wise convolution | canonical convolution | canonical convolution | The DWC layer is more lightweight |
| Number of parameters (given full receptive field) | $D \times L$ | $D \times L \times Z$ | $L \times Z$ | The DWC layer avoids an extra hyper-parameter Z |
| Pooling-demanding | ✗ | ✓ | ✓ | The DWC layer does not incorporate pooling and preserves the ordering information |
| Padding | ✓ | ✗ | ✗ | The DWC layer incorporates a padding operation to ensure its input and output have the same shape, enabling residual link |
| Receptive field | large | limited | L | The DWC layer controls the risk of overfitting and thus makes it feasible to incorporate large receptive field |
| Meta-former architecture | ✓ | ✗ | ✗ | The DWC layer adapts the meta-former architecture. Notably, given meta-former architecture, the DWC layer remains superior than other token-mixers, see section 5.3 for reference. |
| Complexity (accelerated) | $\mathcal{O}(D \times L \log L)$ | $\mathcal{O}(Z \times L \log L \times D \log D)$ | $\mathcal{O}(ZD \times L \log L)$ | The DWC layer performs 1-D convolution in each channel, thus can be accelerated with 1D FFT; the horizontal convolution performs 2-D convolution and employs Z kernels, thus can be accelerated with 2D FFT with larger complexity. |

Table E10: MLP vs DWC with Full Receptive Field.

| Technical difference | DWC layer (FRF) | MLP | Advantage of the DWC layer |
|---|---|---|---|
| Number of parameters | $D \times L$ because each hidden dimension has unique convolution weight. | $L \times L \times D$ with unique MLP per hidden dimension, $L \times L$ with shared MLP across dimensions. | The number of parameters in a DWC layer is comparatively lower than that of an MLP, particularly when dealing with longer sequences. |
| Meta-former architecture | ✓ | ✗ | The DWC layer is based on the Meta-Former architecture, which distinguishes it from canonical MLPs. |
| Accelerable with FFT | ✓ | ✗ | The FFT can accelerate the computation of the DWC layer, but it does not speed up the computation of MLP. |

necessary. Both strategies cooperate to effectively minimize training error and control the generalization gap, consequently reducing the generalization error $E_{\text{out}}$.

# E    MORE COMPARISON WITH EXISTING METHODS

## E.1    COMPARISON WITH MLPS

A potential concern might arise regarding the Depth-Wise Convolution (DWC) layer reducing to a Multi-Layer Perceptron (MLP) under a full receptive field (FRF) setting. However, key differences exist between the two, as outlined in Table E10. To highlight the core differences:

- Acceleration: FFT can expedite the computation of DWC layers, a feature not applicable to MLPs. This underscores a fundamental difference in computational efficiency.

- Dimensions: For a one-dimensional input sequence $X \in \mathbb{R}^{L \times 1}$, an MLP requires a weight matrix $W \in \mathbb{R}^{L \times L}$ to maintain output shape (for residual link). In contrast, a DWC layer with FRF only needs $W \in \mathbb{R}^{K \times 1}$ with K = L, highlighting its lightweight nature even in the FRF setting.

ConvFormer is a non-trivial advancement in the field of sequential user modeling albeit with a relatively simple architecture. This is evidenced by the observation that it is not feasible to obtain ConvFormer by simply adding a simple update to existing sequential user models. The details are formulated as follows.

Table F11: Performance on four datasets. Bold and underlined fonts indicate the best and second-best result, respectively. "*" marks the significant improvement over the second-best result with p-value < 0.01 on the one-sample t-test.

| Dataset | Metric | FM | AutoInt | GRU4Rec | Caser | SASRec | BERT4Rec | GCSAN | FMLP | L-Mixer |
|---------|--------|-----|---------|---------|-------|--------|----------|-------|------|---------|
| Beauty | H@1 | 0.0405 | 0.0447 | 0.1337 | 0.1337 | 0.1870 | 0.1531 | 0.1973 | 0.2011 | **0.2020** |
| | H@5 | 0.1461 | 0.1705 | 0.3125 | 0.3032 | 0.3741 | 0.3640 | 0.3678 | 0.4025 | **0.4151***  |
| | N@5 | 0.0934 | 0.1063 | 0.2268 | 0.2219 | 0.2848 | 0.2622 | 0.2864 | 0.3070 | **0.3143*** |
| | H@10 | 0.2311 | 0.2872 | 0.4106 | 0.3942 | 0.4696 | 0.4739 | 0.4542 | 0.4998 | **0.5139*** |
| | N@10 | 0.1207 | 0.1440 | 0.2584 | 0.2512 | 0.3156 | 0.2975 | 0.3143 | 0.3385 | **0.3462*** |
| | MRR | 0.1096 | 0.1226 | 0.2308 | 0.2263 | 0.2852 | 0.2614 | 0.2882 | 0.3051 | **0.3105*** |
| Sports | H@1 | 0.0489 | 0.0644 | 0.1160 | 0.1135 | 0.1455 | 0.1255 | 0.1669 | 0.1646 | **0.1652** |
| | H@5 | 0.1603 | 0.1982 | 0.3055 | 0.2866 | 0.3466 | 0.3375 | 0.3588 | 0.3803 | **0.3919*** |
| | N@5 | 0.1048 | 0.1316 | 0.2126 | 0.2020 | 0.2497 | 0.2341 | 0.2658 | 0.2760 | **0.2823*** |
| | H@10 | 0.2491 | 0.2967 | 0.4299 | 0.4014 | 0.4622 | 0.4722 | 0.4737 | 0.5059 | **0.5150*** |
| | N@10 | 0.1334 | 0.1633 | 0.2527 | 0.2390 | 0.2869 | 0.2775 | 0.3029 | 0.3165 | **0.3221*** |
| | MRR | 0.1202 | 0.1435 | 0.2191 | 0.2100 | 0.2520 | 0.2378 | 0.2691 | 0.2763 | **0.2804*** |
| Toys | H@1 | 0.0257 | 0.0448 | 0.0997 | 0.1114 | 0.1878 | 0.1262 | 0.1996 | 0.1935 | **0.1984** |
| | H@5 | 0.0978 | 0.1471 | 0.2795 | 0.2614 | 0.3682 | 0.3344 | 0.3613 | **0.4063** | 0.4052 |
| | N@5 | 0.0614 | 0.0960 | 0.1919 | 0.1885 | 0.2820 | 0.2327 | 0.2836 | 0.3046 | **0.3068** |
| | H@10 | 0.1715 | 0.2369 | 0.3896 | 0.3540 | 0.4663 | 0.4493 | 0.4509 | **0.5062** | 0.5053 |
| | N@10 | 0.0850 | 0.1248 | 0.2274 | 0.2183 | 0.3136 | 0.2698 | 0.3125 | 0.3368 | **0.3391** |
| | MRR | 0.0819 | 0.1131 | 0.1973 | 0.1967 | 0.2842 | 0.2338 | 0.2871 | 0.3012 | **0.3043** |
| Yelp | H@1 | 0.0624 | 0.0731 | 0.2053 | 0.2188 | 0.2375 | 0.2405 | 0.2493 | 0.2727 | **0.2797** |
| | H@5 | 0.2036 | 0.2249 | 0.5437 | 0.5111 | 0.5745 | 0.5976 | 0.5725 | 0.6191 | **0.6260*** |
| | N@5 | 0.1333 | 0.1501 | 0.3784 | 0.3696 | 0.4113 | 0.4252 | 0.4162 | 0.4527 | **0.4602*** |
| | H@10 | 0.3153 | 0.3367 | 0.7265 | 0.6661 | 0.7373 | 0.7597 | 0.7371 | 0.7720 | **0.7721** |
| | N@10 | 0.1692 | 0.1860 | 0.4375 | 0.4198 | 0.4642 | 0.4778 | 0.4696 | 0.5024 | **0.5077*** |
| | MRR | 0.1470 | 0.1616 | 0.3630 | 0.3595 | 0.3927 | 0.4026 | 0.4006 | 0.4299 | **0.4360*** |

## E.2 COMPARISON WITH CURRENT CNN-BASED SOLUTIONS

There are several major differences between ConvFormer and the conventional convolution networks, particularly Caser (Tang & Wang, 2018), an exemplar CNN-based sequential user model. ConvFormer incorporates a meta-former architecture with a residual link, layer normalization, and disentanglement between the token-mixer and the channel-mixer, which makes it fundamentally different from canonical convolution networks[6]. Besides, the DWC layer in ConvFormer is fundamentally different from the horizontal and vertical convolution layers proposed by Tang & Wang (2018) which are commonly used in sequential user models. The technical differences and advantages of the DWC layer are illustrated in Table E9. To summarize:

- The horizontal convolution (Tang & Wang, 2018) is essentially a canonical convolution followed by a max-pooling layer, and even with a large receptive field, it is not equivalent to our DWC layer. In fact, the horizontal convolution layer is similar to the Conv-V variant in Section 5.3.2. According to Figure 5, the Conv-V variant is obviously inferior to the DWC layer, which is a reasonable result since it violates our criteria of lightweight architecture.

- The vertical convolution is actually a canonical multi-kernel 2D convolution with kernel shape $1 \times L$, cascaded by a max-pooling layer. In contrast, the DWC layer is a depth-wise single-kernel 2D convolution with kernel shape $D \times L$, free of a cascaded pooling layer.

## F BROADER IMPACT

ConvFormer, as a practical implementation of our proposed criteria, demonstrates superior performance over various existing methods in sequential user modeling. To further validate the generality and applicability of these criteria, we introduce another model, L-Mixer. It uses an affine layer acting as the token mixer, which is embarrassingly simple yet satisfies the proposed criteria simultaneously. The performance outcomes of L-Mixer, detailed in Table F11, show promise with respect to current art model. However, it lacks adaptability to varying input lengths during inference and cannot be

---

[6]The incorporation of meta-former architecture is a non-trivial technical point that has recently received attention from the machine learning community (Yu et al., 2022).

accelerated like the LighTCN layer. Hence, ConvFormer remains our primary exemplar and focus in the main text.

We hope that the leading performance of these two simple yet effective modifications will inspire future research in this domain. These models demonstrate that adherence to well-thought-out design criteria can lead to efficient and effective solutions in sequential user modeling. Future research could explore further refinements to these models, address their limitations, and perhaps even develop new approaches inspired by the principles laid out in our study.

