# OpenReview forum: "ConvFormer: Revisiting Token-mixers for Sequential User Modeling"
_ICLR.cc/2024/Conference — Submitted to ICLR 2024_

### Official Review · Reviewer_TxzE · 2023-10-30

**Soundness:** 2 fair
**Presentation:** 2 fair
**Contribution:** 3 good
**Rating:** 5
**Confidence:** 5

**Summary:**

The paper challenges the conventional approaches surrounding token mixers in sequential user modeling. It highlightes that the self-attentive token mixer of the Transformer architecture is outperformed by simpler strategies, particularly in this domain. The authors identify and emphasize three key empirically validated criteria for effective token mixers - order sensitivity, a large receptive field, and a lightweight architecture. Based on that, they introduce ConvFormer, a straightforward yet effective modification to the Transformer. The approach combines depth-wise and channel-wise convolution layers, which yields performance improvements. The results offer insights into the design principles that enhance the efficacy of token mixers in the realm of recommender systems.

The paper is generally well-structured, it contains rich literature survey and a lot of experimental results. However, it is not always easy to read. While all parts of the text are logically connected, the content within each part often refers the reader across the entire text and even into the appendix. This is especially critical in the introductory parts where the authors outline their ideas and try to convey the main message. This also leads to several major issues that are explained in more detail below.

**Strengths:**

- the topic of finding alternatives to self-attention mechanism is interesting and important, the paper can be viewed as a good overview of the related literature
- the work contains a lot of empirical studies that allow greater granularity in analyzing the sequential learning process
- the idea of replacing self-attention with convolutional mechanisms has the potential to become a valuable contribution into the field

**Weaknesses:**

- the description of the main experiments outlining key insights of the paper lack clarity
- the design of the experiments and the methodology seem contradictory to the main claims of the paper
- some straightforward baselines are missing from comparisons, which makes conclusion on superiority less convincing

**Questions:**

### **Causality**
The most obscure part of the work is the experimental setting for devising the key three principles pertinent to the construction of effective and efficient sequential recommenders. Section 3 is devoted to this description and deserves more accurate wording and rich explanations. For some reason, there are no illustrations provided in Section 3 for the considered SAR-O, SAR-P, and SAR-R models. There is a footnote referring to the appendix A.4 section, which also doesn't contain the figures. The most critical part that is missing in the explanation of these models is how the sequential order is taken into account and how the causal structure of learning is preserved (so that the leaks from future are avoided). For example, there are two contradicting statements in the description of the SAR-O approach: the authors claim that it's independent of the input sequence and then one sentence later comes "SAR-O is directly sensitive to the order of the input items". The same problem goes for SAR-P and SAR-R models. In the case of SAR-P each input element for the sequence is handled independently via MLP, how can it directly depend on the order? In the case of SAR-R, the item-to-item mixing matrix is random, so how is it "order-sensitive"? Explanation probably requires more than a few sentences. Graphical representation along with mathematical formulation in this section would help to convince the reader and minimize confusion.

Having a more coherent explanation is especially critical for supporting the corresponding claims on superiority of those simpler mixers over the original SASRec model. Moreover, there's another issue related to these claims. The authors use a simple leave-last-out holdout sampling scheme to hide test items, which is subject to "recommendations from future" type of leaks (see [Ji at el 2023]). Combined with non-causal item-to-item mixer this would create an unfair advantage to the proposed modifications of the SASRec, which employs strictly triangular mask and ensures causal learning. So, seeing the superiority would be no surprise in this case, but it would not address the sequential learning task anymore. Hence, modifying the experimentation scheme that excludes the "recommendations from future" leaks is particularly important as well. One can also run a simple pre-check by lifting the triangular mask from SASRec. Most likely it will outperform the proposed modifications, which will also reveal the general problem with the setup.

The related issue arises with receptive field size experiments in Section 3.2. When K nearest neighbors are selected, the simple symmetric distance is used |i-j|\<K. So, no causal mask is used here either, which only reaffirms suspicions regarding the possible leaks in the training.

There also seems to be a contradiction with the ways SASRec is represented in the text.  In section 4, the authors outline three key components of an effective sequential learner and state that SASRec does not correspond to the two of them: order sensitivity and large receptive field. However, both properties are ingrained into the sequential self-attention mechanism: it does preserve information on the order via successive learning of hidden states over the sequence via triangular mask (positional embeddings also count but their contribution is marginal), and it operates over all preceding items at any step within an input sequence. Maybe something different was meant by the authors, but it requires careful wording and more clarification.

Continuing on the topic of information leakage, the proposed solution also raises concerns. Specifically, the depth-wise convolution that operates within each hidden component, meaning it blends in information across entire sequence, and therefore loses the causal structure and allows capturing information from future elements. A careful analysis of the leaks would help resolve the concerns.

### **Baselines**
Even though the authors claim to only modify the attention mechanism, the proposed solution  also deviates from the original SASRec implementation in terms of the loss function. Changing the loss function can have a significant impact on the quality of SASRec even without changing its architecture, see e.g. [Klenitskiy and Vasilev 2023], [Frolov et al. 2023], [Petrov and Macdonald 2023].
So, the comparison of the proposed approach (as an alternative to SASRec) is unfair unless either the same loss is used in the proposed model or the SASRec's loss function is also modified to employ the proposed pairwise objective. Otherwise, the claimed contribution of the architectural changes is not convincing.

### **Other issues**
The authors mention computational cost of long sequences, however, most of the real applications in recommender systems have relatively short sequences (comparing to e.g. NLP applications). Having the table with datasets statistics including average sequence lengths (like the one in the SASRec paper) would help to see that. Even length 50 is rarely attainable in most datasets. Applying FFT to compute convolutions is a logical step but it is only remotely related to the problem of long sequences. It is ubiquitously used to compute convolutions and should not be presented as a particular contribution by the authors.

### **References**

Ji Y, Sun A, Zhang J, Li C. A critical study on data leakage in recommender system offline evaluation. ACM Transactions on Information Systems. 2023 Feb 7;41(3):1-27.

Klenitskiy, A. and Vasilev, A., 2023, September. Turning Dross Into Gold Loss: is BERT4Rec really better than SASRec?. In _Proceedings of the 17th ACM Conference on Recommender Systems_ (pp. 1120-1125).

Frolov E., Bashaeva L., Mirvakhabova L., Oseledets I. Hyperbolic Embeddings in Sequential Self-Attention for Improved Next-Item Recommendations. Under review at https://openreview.net/forum?id=0TZs6WOs16.

Petrov, A.V. and Macdonald, C., 2023, September. gSASRec: Reducing Overconfidence in Sequential Recommendation Trained with Negative Sampling. In _Proceedings of the 17th ACM Conference on Recommender Systems_ (pp. 116-128).

---

> ### Author Response · Authors · 2023-11-18
> **Response to Reviewer TxzE [1/4]**
>
> Thank you very much for your thoughtful and comprehensive feedback, which is immensely helpful in improving the quality and clarity of our work.
> We find that the weaknesses you pointed out are well manifested as the specific queries in the question section. Therefore, we opt to answer the questions directly.
>
> **[Q1.1] Causality. On the setting for key principles. Section 3 is devoted to this description and deserves more accurate wording and rich explanations. For some reason, there are no illustrations provided in Section 3 for the considered SAR-O, SAR-P, and SAR-R models. There is a footnote referring to the appendix A.4 section, which also doesn't contain the figures...Explanation probably requires more than a few sentences. Graphical representation along with mathematical formulation in this section would help to convince the reader and minimize confusion.**
> - **More Explanations with Graphical Illustrations.** Thank you for your kind prompt. We have included comprehensive graphical illustrations and mathematical formulations in **Appendix C.2** for SAR and its variants. The corresponding footnote in the main text has been updated to direct readers to these contents.
> - **Clarifying Order-Sensitivity in SAR and Its Variants:** Recognizing the importance of elucidating the order-sensitivity of SAR variants, **we have expanded our descriptions in the revised Appendix C.2**.
> The concept of order sensitivity in a token mixer can be defined as follows: for any item in the input sequence, changing the order of other items alters the representation outputted by the token mixer. To illustrate this, consider the representation of the $i$-th item in the input sequence, denoted as $r_i \in \mathbb{R}^\mathrm{D}$.
> In SAR and its variants, the output representation of $r_i$ is given by $s_i=\sum a_{i,j}r_j$ (we omit the value mapping here for brevity. It is reasonable since the value mapping is conducted on each step isolatedly, without impact on order sensitivity).
> If we switch the order of two items, $j_1$ and $j_2$, the output representation changes accordingly: $s_i^\prime=\sum_{j=1,j\neq{j_1},j\neq j_2}^{\mathrm{L}}a_{i,j}r_j + a_{i,j_1}r_{j_2}+a_{i,j_2}r_{j_1}$, reflecting the model's sensitivity to the item order. **In SAR**, $s_i$ does not exhibit order sensitivity.
>     This is because the attention weights are dynamically determined based on the similarity between items, irrespective of their positions in the sequence. Consequently, swapping two items, $j_1$ and $j_2$, does not alter the output representation of the $i$-th item, i.e., $s^\prime_i=s_i$.
>     **In SAR-O and SAR-R**, the weights $a_{i,j}$ are fixed with respect to positions rather than dynamically calculated, which makes $s_i$ sensitive to the order of items, manifested as $s^\prime_i \neq s_i$.
>     **In SAR-P**, the only difference from SAR-O is that the weights $a_i$ are dependent on $r_i$: a_i=[a_{i,j}]_{j=1:L}=\mathrm{MLP}(r_i). Since the generated weights remain dependent on the position index $j$, $s{^\prime}_i\neq s_i$ holds and thus SAR-P is order-sensitive.

---

> ### Author Response · Authors · 2023-11-18
> **Response to Reviewer TxzE [2/4]**
>
> **[Q1.2] Causality. Leave-last-out holdout sampling scheme combining with lacking causal mask would create an unfair advantage ... modifying the experimentation scheme that excludes the "recommendations from future" leaks is particularly important as well.**
>
> Thank you for your valuable comment!  We highly value any non-clarity here and are pleased to give our response at a structral manner.
>
> - **Preserving of Causality in Our Study.**  In our methodology, we have incorporated causality using an equivalent approach to a traditional causal mask. For an input sequence like [1,2,3,4,5], we employ autoregressive generation of sub-sequences ([1], [1,2], [1,2,3], [1,2,3,4]) to predict the next item ([2], [3], [4], [5]) respectively, minimizing the prediction error, as demonstrated in our code (lines 11-19 in datasets.py). It effectively mirrors the function of an explicit causal mask (i.e., multiplying the attention matrix with a causal mask and then minimize the error to predict [2,3,4,5] using the entire input sequence) and avoids the concerned leakage. Therefore, we do not confer any unfair advantage to the proposed SAR variants and ConvFormer in terms of causality. **We have highlighted it in the revised section 4.2.3.**
>
> - After handling the raised concern "leave-last-out scheme coupling with non-causal item-to-item mixer would create an unfair advantage to SAR variants", we turn to discuss the leave-last-out scheme itself. In comparison to another popular scheme, the random split [1], the leave-last-out scheme has been more effective in mitigating leakage. We agree that in some instances, this scheme also suffers from data leakage. However, the critism [1] mainly focuses on collaborative filtering that recommends based on **user-wise similarity**. The root (see Figure 5 in [1]) is that the future data would be falsely involved when calculating the similarity between user A, B, C which interacted with the recommender chronologically with overlapping. In this case, the "future" interactions of C would be leaked to A by calculating the similarity between A-B and B-C.
> In contrast, sequential recommendation is less prone to such root of leakage (i.e., user-wise similarity), since it models each user individually and avoids user-wise similarity calculations.
> Furthermore, we slightly note that in the realm of recommendation, **each split scheme has its trade-offs**; therefore, the key is to align the scheme with business demanding and respect it for fair comparison. The timepoint split scheme suggested by [1], while avoiding certain biases, leads to shorter and incomplete test sequences that misalign with real-world serving scenarios. In comparison, the leave-last-out scheme aligns more with the actual serving scenarios, where using the entire (long) user history for recommendations is common, especially in mature business. This decision was also influenced by our industrial experience and the need to stay consistent with established public benchmarks that sometimes lack timestamp data.
>
>
> **[Q1.3] Causality. One can also run a simple pre-check by lifting the triangular mask from SASRec. Most likely it will outperform the proposed modifications, which will also reveal the general problem with the setup.**
>
> Thank you for your sincere comment. We are pleased to clarify any concerns here.
> - **Lifting the causal mask from SASRec is not necessary.** The reason is simple: we have preserved causality in training data generation beforehand, which mirrors the function of causal mask. See the response to Q1.2 for details.
> - **Empirical Evidence.** To further validate our stance, we conducted additional empirical studies following your kind suggestion.
> Specifically, we autoregressively generate training sub-sequences. Then, we enable and disable the causal mask. The experiments are conducted using 5 seeds. Due to changes in the hardware platform, experimental results may exhibit minor variations. The results below show that (1) the presence or absence of the causal mask does not significantly affect SASRec's performance; (2) both implementations with and without causal mask did not outperform our simpler alternatives. Therefore, **lifting the causal mask does not invalidate our associated claim in the associated section**: `Simpler, order-sensitive alternatives to the
> item-to-item token mixer in SAR leads to little performance drop.`
>
> | Data | Causal Mask | HIT@1 | HIT@5 | HIT@10 | NDCG@5 | NDCG@10 | MRR |
> |---:|---:|---|---:|---:|---:|---:|---:|
> | Yelp | √ | 0.22566±0.00199 | 0.56924±0.00292 | 0.74524±0.00270 | 0.40226±0.00237 | 0.45932±0.00232 | 0.38448±0.00207 |
> | Yelp | × | 0.22598±0.00163 | 0.56780±0.00368 | 0.74372±0.00153 | 0.40166±0.00257 | 0.45874±0.00185 | 0.38418±0.00181 |
> | Sports | √ | 0.14644±0.00230 | 0.34644±0.00315 | 0.46870±0.00116 | 0.24934±0.0028 | 0.28870±0.00218 | 0.25264±0.00237 |
> | Sports | × | 0.14494±0.00240 | 0.34560±0.00159 | 0.46448±0.00184 | 0.24832±0.00191 | 0.28660±0.00162 | 0.25124±0.00189 |

---

> ### Author Response · Authors · 2023-11-18
> **Response to Reviewer TxzE [3/4]**
>
> **[Q1.4] Causality. The related issue arises with receptive field size experiments in Section 3.2. When K nearest neighbors are selected, the simple symmetric distance is used |i-j|<K. So, no causal mask is used here either, which only reaffirms suspicions regarding the possible leaks in the training.**
> - Thank you for your sincere comment. We believe **this concern could be resolved after rectifying Q1.2.** Specifically, causality is ensured by autoregressively generating sub-sequences for training beforehand, which is equivalent to using causal mask for the entire sequence.
>
> **[Q1.5] Causality. There seems to be a contradiction. In section 4, the authors outline three key components and state that SASRec does not correspond to the two of them: order sensitivity and large receptive field. However, both properties are ingrained into the sequential self-attention mechanism...it operates over all preceding items at any step within an input sequence.**
>
> - **Large Receptive Field.** We acknowledge the importance of a large receptive field in SASRec (rather than opposing it), and this is consistent with our perspective in the main text. We have highlighted the role of a large receptive field as a key factor in SASRec's performance: `By deconstructing vari
> ous aspects of self-attentive token mixer, we identify two factors favouring its performance: a large
> receptive field and a lightweight architecture.`, and `These findings underscore the importance of a large receptive field for SAR's performance.`
> If there are any inaccuracies or potentially misleading statements in our manuscript regarding this point, we welcome specific references to them, and we are committed to rectifying any such issues.
> - **Order Sensitivity.**  As previously discussed in response to Q1.2, we have ensured causality through autoregressive generation of training sequences, an approach that mirrors the function of a causal mask. In each generated sub-sequence, SASRec does not exhibit order sensitivity, as evidenced by the relationship $\mathbf{r}{^\prime}_i=\mathbf{r}_i$ (detailed in the resposne to Q1.1). As an equivalent (but kind of difficult to interpret) view, we can use causal mask to establish causality and then minimizing the prediction error from the 2nd to the L-th items in the sequence. It in essence devides the sequence [1,2,3,4,5] into L-1 sequences: [1], [1,2], [1,2,3], [1,2,3,4] for predicting the next item; in each subsequence SASRec does not exhibit order sensitivity.
>
>
> **[Q1.6] Causality. Continuing on the topic of information leakage, the proposed solution also raises concerns. Specifically, the depth-wise convolution that operates within each hidden component, meaning it blends in information across entire sequence, and therefore loses the causal structure and allows capturing information from future elements. A careful analysis of the leaks would help resolve the concerns.**
> - Thank you for your sincere comment. We believe **this issue has been effectively addressed after rectifying Q1.2.** The **causality has been ensured by autoregressively generating training sequences beforehand**, which is equivalent to exerting causal attention mask and optimize the model outputs at multiple steps simultaneously. It also prevents depth-wise convolution from introducing leakage from future elements.
>
>
> **[Q2] Baselines. To make comparison fair, the same loss should used in the proposed model or the SASRec's loss function is also modified to employ the proposed pairwise objective.**
>
> - **The loss functions of SASRec and ConvFormer are the same.** We acknowledge and appreciate your suggestion regarding the consistency of loss in model comparison. We apologize for an unexpected typo in the initial manuscript regarding the loss function formulation. In fact, they **share the same loss calculation**. As evidenced in our uploaded code, both SASRec and ConvFormer, along with all toy models, are trained using the same PairwiseTrainer (referenced in line 173 of src/trainers.py) and employ the same bce loss calculator (mentioned in line 152 of src/trainers.py).
>
> - **Highlighting Consistency in Revised Documentation:** In the revised version of our paper, we have corrected the formulation of loss function. Then, we noted in the footnote 1 to highlight experimental fairness: `We use the experimental settings in Section 5.1, e.g., setting the embedding size to 64, the maximum
> sequence length to 50. Experiments are repeated 5 times with different seeds. We
> only modify the token mixer matrix A in (1), maintaining learning objectives and tricks identical to SASRec.`

---

> ### Author Response · Authors · 2023-11-18
> **Response to Reviewer TxzE [4/4]**
>
> **[Q3.1] Other issues. Most of the real applications in recommender systems have relatively short sequences.**
> - **Variability in Optimal Sequence Lengths:** The optimum of sequence length (L) varies based on business needs and dataset characteristics. For instance, movie-lens (used in our appendix A.3) exhibits optimal performance at L=200 as reported by BERT4Rec [8]. Additionally, in many practices, particularly in industry, behavior sequences can be considerably longer (L>1,000), since incorporating more user behaviors shows consistent performance improvements (referenced in Fig.1 in [9]). It highlights the efficiency and efficacy of handling long-term user sequences, sometimes extending to **extreme lengths (L>54,000) and life-long sequences** [9,10,11,12]. **Therefore, the complexity of processing long input sequences becomes a critical aspect when designing sequential user models.** Motivated by this, we extended our complexity discussion to long sequences, to avoid associated concerns regarding ConvFormer.
>
>
> **[Q3.2] Other issues. Applying FFT to compute convolutions is a logical step but it is only remotely related to the problem of long sequences. It is ubiquitously used to compute convolutions and should not be presented as a particular contribution by the authors.**
> - **Clarification of FFT's Role in the Study:** According to the response to Q3.1, the user sequence would be quite lengthy. It's important to clarify that **FFT is employed in our study not as a central novelty, but as a strategic component**  to manage the complexity of lengthy user sequences, to bypass unexpected concerns.  Despite these technical points, **our primary innovation and contribution is investigating and rethinking token mixer design in sequential user modeling**. It yields empirical rules that flourish a simple model achieving SOTA performance and could influence future innovations. To highlight the main contribution, we have only briefly mentioned the complexity issue and the FFT solution in the main text, allocating only a small section for this, while detailed descriptions, algorithms, and corresponding experiments are included in the appendix.
>
> - **Comparison with Recent FFT-empowered Works:**
> Many recent studies [2-4] and contributions to ICLR 2024 [5-7] are featured by the use of Fourier analysis and Fourier convolution in specific problem-solving contexts. In contrast, our work does not highlight the application of Fourier analysis as its main novelty. Our technical novelty is the overall framework (as shown in the second contribution in our introduction), which integrates the lightTCN module and the meta-former architecture. The use of FFT for convolution is to support these components by addressing computational complexities, not as a standalone innovation.
>
> ### **Reference**
>
> [1] Ji, Yitong, et al. "A critical study on data leakage in recommender system offline evaluation." ACM Transactions on Information Systems 41.3 (2023): 1-27.
>
> [2] Li, Chongyi, et al. "Embedding Fourier for Ultra-High-Definition Low-Light Image Enhancement." The Eleventh International Conference on Learning Representations. 2022.
>
> [3]Zhou, Man, et al. "Fourmer: an efficient global modeling paradigm for image restoration." International Conference on Machine Learning. PMLR, 2023.
>
> [4]Huang, Zhipeng, et al. "Adaptive Frequency Filters As Efficient Global Token Mixers." Proceedings of the IEEE/CVF International Conference on Computer Vision. 2023.
>
> [5] "Revitalizing Channel-dimension Fourier Transform for Image Enhancement." Submission to ICLR, Openreview, 2023.
>
> [6] "HoloNets: Spectral Convolutions do extend to Directed Graphs." Submission to ICLR, Openreview, 2023.
>
> [7] "On the Matrix Form of the Quaternion Fourier Transform and Quaternion Convolution." Submission to ICLR, Openreview, 2023.
>
> [8] Sun, Fei, et al. "BERT4Rec: Sequential recommendation with bidirectional encoder representations from transformer." Proceedings of the 28th ACM international conference on information and knowledge management. 2019.
>
> [9] Pi, Qi, et al. "Practice on long sequential user behavior modeling for click-through rate prediction." Proceedings of the 25th ACM SIGKDD International Conference on Knowledge Discovery & Data Mining. 2019.
>
> [10] Pi, Qi, et al. "Search-based user interest modeling with lifelong sequential behavior data for click-through rate prediction." Proceedings of the 29th ACM International Conference on Information & Knowledge Management. 2020.
>
> [11] Zhang, Yuren, et al. "Clustering based behavior sampling with long sequential data for CTR prediction." Proceedings of the 45th International ACM SIGIR Conference on Research and Development in Information Retrieval. 2022.
>
> [12] Cao, Yue, et al. "Sampling is all you need on modeling long-term user behaviors for CTR prediction." Proceedings of the 31st ACM International Conference on Information & Knowledge Management. 2022.

---

> ### Author Response · Authors · 2023-11-22
> **Thank you for your insightful feedback.**
>
> Thank you very much for your informative and insightful feedback. We understand that the concerns are mainly the preservation of causality and the consistency of loss function. We acknowledge that there were some inaccuracies and omissions in our previous statements, and we apologize for any misunderstanding this may have caused. Thank you for pointing them out. We have included detailed explanations in our revised manuscript (highlighted in red for easy identification) and supplied necessary empirical evidence in our responses.  These clarifications are helpful to handle the corresponding concerns and enhance the coherence of our work.
>
> With the discussion deadline approaching, we are anticipating any further advice, inquiries or remaining concerns you may have. We will make our best effort to handle them.

---

> ### Author Response · Authors · 2023-11-28
> **We find a parallel work as a proof-of-concept for your reference.**
>
> Dear Reviewer TxzE,
>
> Thank you for your comprehensive initial review. We understand that the concerns are mainly the preservation of causality and the consistency of loss function. We have included detailed explanations in our revised manuscript (highlighted in red) and supplied necessary empirical evidence. **We are hopeful to know whether these responses and revisions have handled the concerns raised.**
>
> Recently, we came across **a parallel submission** [1], which bears relevance to our work and could mitigate **your main concerns regarding causality.** Overall, ModernTCN [1] is featured by a modernTCN layer with applications to sequence modeling, closely **aligning with our LighTCN block in terms of technical aspects**.  In comparison, **we additionally preserve causality** by autoregressively generating sub-sequence for training (see our initial response to Q1.2 for details).  The community’s positive reception of "ModernTCN" (reflected in its scores of 8-8-8) could **corroborate the practical efficacy of our LighTCN block and the proposed criteria in broad fields**.
>
>
> Notably, the **research problems** of the two works are different. The parallel work [1] focuses on the problem `Can CNN outperform TransFormer in sequential modeling?` and is featured by a ModernTCN block. **In comparison, our work’s innovation and contribution extend beyond the specific LighTCN block**. We delve into a critical reassessment of the current trend favoring simpler methods over Transformer-style approaches.
> In doing so, we not only investigated the research problem:  `Can CNN outperform TransFormer in sequential modeling?`, but also went further to explore `Are there common criteria for effective token-mixers in sequential modeling?` This exploration has led to the formulation of three pivotal criteria. Adhering to the criteria, simple token mixers (e.g., even an affine layer in Appendix F) can achieve leading performance. **The significance of our work, therefore, lies in broader implications for architectural methodology in sequential user modeling, offering key insights for future innovations, rather than proposing a specific CNN-like block with SOTA performance.**
>
> [1] "ModernTCN: A Modern Pure Convolution Structure for General Time Series Analysis." https://openreview.net/forum?id=vpJMJerXHU
>
> We are hopeful that our response and the review of [1] may encourage you to reconsider your assessment of our work. In light of the contrasting viewpoints in a recent emergent review and an existence of a parallel work, such an engagement becomes particularly important.
> With five days remaining for discussion, we are fully committed to engaging in any further technical dialogues you may have.

---

### Official Review · Reviewer_DKuE · 2023-11-01

**Soundness:** 4 excellent
**Presentation:** 4 excellent
**Contribution:** 3 good
**Rating:** 6
**Confidence:** 4

**Summary:**

This paper summarize the key ingredient to build a good user modeling architecture. The three components are: 1) order sensitive, 2) large receptive field 3) Light weight. They provide experiment to prove their argument. They propose ConvFormer with the fourier transformation and achieves SOTA performance on the offline dataset.

**Strengths:**

1. Identify the problem of self-attentive transformer when it was applied in user modelling
2. Good performance

**Weaknesses:**

No online real experiment
Only the ID feature was considered

**Questions:**

1. What's the relationship between the optimal K (kernel size) and the dataset sequence length?
2. How would you choose the input sequence length?
3. Will adding more block improve performance?

---

> ### Author Response · Authors · 2023-11-18
> **Response to Reviewer DKuE [1/4]**
>
> Thank you for your kind words and recognition of the soundness, presentation, and contributions of our work. We deeply appreciate the critical insights and the interesting aspects pointed out in your comments, which we believe are valuable for enhancing the depth and breadth of our research. The suggestions for further empirical investigation are particularly interesting, which could improve the quality of this work. We are therefore pleased to undertake additional empirical studies that can shed more light on the aspects highlighted in your feedback.
>
> **[W1] No online real experiment. Only the ID feature was considered.**
>
> - **Online Experiments.** We appreciate your emphasis on real-world application validity. This work follows the evaluation protocol of current arts in the field [1-4]. **Recognizing the importance of real-world applications, we have implemented ConvFormer on our private business dataset, which has surpassed most existing protocols [1-4]**. While our results show promise in both open benchmarks and industrial data, in our industry, new features typically undergo extensive offline testing before being integrated into online systems. As academic researchers collaborating with the business sector, we lack the authority to independently initiate online deployment of the model.
>
> - **Non-ID Features:** We agree that incorporating non-id features is a good supplementary to this work. However, it's important to note that **the majority of academic benchmarks (and our industrial dataset) primarily focus on ID features.**
> Given that additional information can be integrated through supplementary representation extractors – a process not central to the core of sequential user modeling – most studies in this area concentrate solely on item IDs [1,2,3].
> As a research paper, we adhere to this established norm of sequential user modeling, which typically involves using historical item IDs to predict future item IDs.

---

> ### Author Response · Authors · 2023-11-18
> **Response to Reviewer DKuE [2/4]**
>
> **[Q1] What's the relationship between the optimal K (kernel size) and the dataset sequence length (L)?**
> - **Optimal K given different L.** Your query regarding the optimal kernel size (K) in relation to the sequence length (L) is insightful and directly pertains to the efficacy of the large receptive field criterion in sequential user modeling. Our experimental investigations on the Yelp dataset, conducted with five random seeds and adhering to the training and evaluation protocols specified in the main text, shed light on this relationship.  Due to changes in the hardware platform, experimental results may exhibit minor variations.
> The experiments indicate **a general trend: as the sequence length increases, so does the value of the optimal kernel size**. This trend suggests that longer sequences benefit from larger kernel sizes, which effectively capture the broader context within the sequence. For instance, on the Yelp dataset, when L is set to 10 and 20, the optimal values of K are observed to be 10 and 20, respectively. Further extending L to 100 results in optimal K values in the range of 70-100. Following your kind suggestion, the efficacy of large receptive field (the criterion 2) is further validated and underscored.

---

> ### Author Response · Authors · 2023-11-18
> **Empirical study: relationship between the optimal K and L on Yelp dataset.**
>
> |L|K|HIT@1|HIT@5|HIT@10|NDCG@5|NDCG@10|MRR|
> |---:|---:|---:|---:|---:|---:|---:|---:|
> |10|3|0.242+0.0041|0.5731+0.0048|0.733+0.0029|0.4131+0.0042|0.465+0.003|0.3948+0.0036|
> |10|5|0.2583+0.0068|0.5934+0.0035|0.7467+0.0005|0.432+0.0052|0.4818+0.0044|0.4115+0.0054|
> |10|10|**0.2632+0.008**|**0.599+0.0043**|**0.7529+0.0016**|**0.4374+0.0063**|**0.4874+0.0053**|**0.4169+0.0063**|
> |20|3|0.2549+0.001|0.5941+0.0028|0.7548+0.002|0.4303+0.0013|0.4824+0.0011|0.4102+0.0007|
> |20|5|0.262+0.0039|0.6095+0.0024|0.7682+0.002|0.4421+0.0034|0.4936+0.0032|0.4201+0.0036|
> |20|10|0.2738+0.0019|0.6208+0.0027|0.7752+0.0016|0.4538+0.0026|0.504+0.0014|0.4309+0.0019|
> |20|20|**0.2738+0.0039**|**0.6209+0.0024**|**0.7755+0.0003**|**0.4545+0.0036**|**0.5047+0.0027**|**0.4317+0.0036**|
> |30|3|0.258+0.0012|0.6008+0.0031|0.76+0.0015|0.4355+0.0027|0.4871+0.0022|0.4145+0.0023|
> |30|5|0.2721+0.0061|0.6197+0.004|0.7741+0.001|0.4525+0.0054|0.5027+0.004|0.4297+0.0053|
> |30|10|**0.2768+0.0034**|**0.6276+0.0052**|0.7804+0.0007|**0.4592+0.0042**|0.5088+0.0025|**0.4353+0.003**|
> |30|20|0.2743+0.0027|0.6257+0.0022|0.7812+0.0014|0.4568+0.0026|0.5074+0.0021|0.4333+0.0026|
> |30|30|0.2754+0.0063|0.6271+0.0057|**0.7834+0.0021**|0.4582+0.0064|**0.5089+0.0052**|0.4344+0.0058|
> |40|3|0.2575+0.0022|0.6003+0.0023|0.7612+0.0017|0.435+0.0019|0.4872+0.0019|0.4143+0.0018|
> |40|5|0.2778+0.001|0.626+0.0036|0.7776+0.002|0.4588+0.0017|0.5081+0.0013|0.4354+0.001|
> |40|10|0.2764+0.007|0.6308+0.0051|0.7844+0.0013|0.4605+0.0059|0.5104+0.0046|0.436+0.0055|
> |40|20|0.2799+0.0036|**0.6311+0.0014**|0.7839+0.0015|0.4624+0.0029|0.512+0.0023|0.4384+0.0031|
> |40|30|**0.2801+0.0045**|0.6308+0.0045|**0.7849+0.0008**|0.4624+0.0045|**0.5125+0.003**|0.4387+0.0039|
> |40|40|0.2797+0.004|0.631+0.0037|0.7841+0.0022|**0.4626+0.0038**|0.5123+0.0022|**0.4387+0.0034**|
> |50|3|0.2596+0.002|0.6043+0.0027|0.7638+0.0018|0.438+0.0022|0.4897+0.0019|0.4166+0.0019|
> |50|5|0.2719+0.0029|0.6222+0.0024|0.7772+0.0016|0.454+0.0028|0.5044+0.0018|0.4307+0.0026|
> |50|10|0.2813+0.0048|0.6271+0.0013|0.7823+0.0026|0.4615+0.0032|0.5119+0.0024|0.4387+0.0034|
> |50|20|0.2759+0.0053|0.6297+0.0039|0.7855+0.0009|0.4599+0.0049|0.5106+0.0036|0.436+0.0046|
> |50|30|0.2802+0.0065|0.6336+0.0024|0.7858+0.0042|0.4639+0.0047|0.5133+0.0026|0.4394+0.0047|
> |50|40|0.2785+0.0027|0.6319+0.0054|**0.7874+0.003**|0.4624+0.0036|0.5129+0.0026|0.4383+0.0027|
> |50|50|**0.2853+0.0025**|**0.6346+0.002**|0.7855+0.0011|**0.4674+0.0021**|**0.5165+0.0012**|**0.4434+0.0017**|
> |60|3|0.2463+0.0023|0.5854+0.0018|0.746+0.0012|0.4218+0.0006|0.4739+0.0007|0.402+0.001|
> |60|5|0.2604+0.0019|0.6063+0.0013|0.7623+0.0001|0.44+0.0011|0.4907+0.0009|0.4179+0.0011|
> |60|10|0.2623+0.0027|0.6115+0.0006|0.7671+0.0013|0.4434+0.0014|0.494+0.0014|0.4207+0.0017|
> |60|20|0.2642+0.0024|0.6115+0.0035|0.7683+0.0013|0.4449+0.0028|0.4958+0.0013|0.4226+0.0021|
> |60|30|0.2663+0.001|0.6139+0.0018|0.7683+0.0014|0.4468+0.0011|0.497+0.0011|0.4242+0.001|
> |60|40|**0.2696+0.0013**|**0.6163+0.0017**|0.7677+0.0009|**0.4499+0.001**|**0.4991+0.0009**|**0.4269+0.0009**|
> |60|50|0.2677+0.0015|0.6144+0.0017|**0.7693+0.0034**|0.448+0.0019|0.4983+0.0003|0.4254+0.0012|
> |60|60|0.2671+0.0024|0.6143+0.0017|0.7682+0.0019|0.4475+0.0021|0.4975+0.0021|0.4248+0.0023|
> |100|3|0.245+0.0011|0.5841+0.0023|0.7444+0.0003|0.4205+0.0018|0.4725+0.0011|0.4008+0.0013|
> |100|5|0.2552+0.0042|0.603+0.0012|0.7636+0.0018|0.4355+0.0017|0.4877+0.0017|0.4139+0.0021|
> |100|10|0.2643+0.0033|0.6115+0.0034|0.7673+0.0019|0.4446+0.0032|0.4951+0.0022|0.4221+0.0026|
> |100|20|0.2662+0.004|0.6158+0.0017|0.7708+0.0023|0.4479+0.0023|0.4983+0.0022|0.4249+0.0026|
> |100|30|0.264+0.0076|0.6131+0.003|0.7696+0.0027|0.4455+0.0052|0.4964+0.0036|0.4229+0.0052|
> |100|40|0.268+0.0022|0.6167+0.0038|0.7708+0.0012|0.4493+0.0032|0.4993+0.0024|0.4262+0.0026|
> |100|50|0.2675+0.0032|0.615+0.0019|0.7715+0.0013|0.4478+0.0012|0.4986+0.0013|0.4252+0.0017|
> |100|60|0.2664+0.002|0.6157+0.0013|0.7731+0.0026|0.4479+0.0019|0.499+0.0007|0.425+0.0016|
> |100|70|**0.2683+0.0004**|0.6161+0.0013|0.7715+0.0007|0.4491+0.0006|**0.4996+0.0004**|**0.4264+0.0003**|
> |100|80|0.2674+0.0039|0.6169+0.0023|0.7722+0.0014|0.4485+0.0029|0.499+0.0022|0.4253+0.0026|
> |100|90|0.2647+0.0041|0.6169+0.0018|**0.7738+0.0014**|0.4479+0.0032|0.4989+0.0026|0.4246+0.0033|
> |100|100|0.2673+0.0045|**0.6179+0.0027**|0.7716+0.0013|**0.4494+0.0038**|0.4993+0.0025|0.4259+0.0036|

---

> ### Author Response · Authors · 2023-11-18
> **Response to Reviewer DKuE [3/4]**
>
> **[Q2] How would you choose the input sequence length?**
>
> - **Common Setting in Academic Studies:** We appreciate your question regarding the selection of input sequence length. In academic contexts, the key is to ensure reproducibility and comparability with existing baselines. Therefore, adhering to the settings of current arts is crucial, i.e., L=50 as demonstrated by references [1,2,4]. We respected and adhered to this setting in our study.
>
> - **Adjustment in Industrial Practice:** L is treated as a tunable hyperparameter in industry, optimized to balance model performance and business requirements for inference speed. Empirically, the optimal sequence length is closely related to the average sequence length of the dataset [2,3]. In our industrial practice, we set L=50, a choice that has been optimized over years and is in line with current online serving requirement.
>
> **[Q3] Will adding more block improve performance?**
>
> - **Number of blocks (N).** This is an insightful comment. Our default setting follows the N=2 configuration from [1].  Nevertheless, we agree the importance of this hyperparameter, and add experiments to assess the impact of varying N. The results from the Yelp dataset, with block numbers ranging from 1 to 4 and two seeds are summarized as follows.
>
>   - **N=1:** Suboptimal performance across all metrics.
>   - **N=2:** Optimal performance, showing notable improvement over N=1.
>   - **N>2:** Little improvement over N=2, suggesting a saturation point at N=2. Although it is expected to observe further improvement, such improvement often entails a finetuning in regularization strength, e.g., increasing dropout, to avoid overfitting. Besides, in sequential user modeling, simple and light models often exhibit superior performance.
>
> | HIT@1  | HIT@5  | HIT@10 | NDCG@5 | NDCG@10 | MRR   | Seed | Number of blocks (N) |
> |--------|--------|--------|--------|---------|-------|------|----------------------|
> | 0.2528 | 0.6037 | 0.7677 | 0.4343 | 0.4876  | 0.4128| 1    | 1                    |
> | 0.2519 | 0.6058 | 0.7692 | 0.4348 | 0.4878  | 0.4124| 2    | 1                    |
> | 0.2843 | 0.6316 | 0.7877 | 0.4655 | 0.5162  | 0.4423| 1    | 2                    |
> | 0.2830 | 0.6357 | 0.7876 | 0.4670 | 0.5163  | 0.4424| 2    | 2                    |
> | 0.2794 | 0.6332 | 0.7862 | 0.4633 | 0.5131  | 0.4387| 1    | 3                    |
> | 0.2853 | 0.6385 | 0.7878 | 0.4692 | 0.5177  | 0.4441| 2    | 3                    |
> | 0.2808 | 0.6320 | 0.7864 | 0.4635 | 0.5137  | 0.4394| 1    | 4                    |
> | 0.2819 | 0.6347 | 0.7847 | 0.4657 | 0.5145  | 0.4408| 2    | 4                    |
>
>
>
>
>
>
> - **Number of Hidden Units (D).** In addition to the number of blocks, we also explored the influence of varying D. Our findings can be summarized as follows.
>
>   - **D=32:** Comparable performance to SOTA models with D=64, but slightly lower.
>   - **D=64:** Balanced and optimal, yielding the best performance.
>   - **D>64:** Increasing D beyond 64 doesn't significantly enhance performance, sometimes even showing a decrease.
>
>
>
> | HIT@1  | HIT@5  | HIT@10 | NDCG@5 | NDCG@10 | MRR   | Seed | Number of hidden units (D) |
> |--------|--------|--------|--------|---------|-------|------|----------------------------|
> | 0.2672 | 0.6342 | 0.7988 | 0.4585 | 0.5120  | 0.4335| 1    | 32                         |
> | 0.2581 | 0.6349 | 0.7970 | 0.4536 | 0.5064  | 0.4267| 2    | 32                         |
> | 0.2843 | 0.6316 | 0.7877 | 0.4655 | 0.5162  | 0.4423| 1    | 64                         |
> | 0.2830 | 0.6357 | 0.7876 | 0.4670 | 0.5163  | 0.4424| 2    | 64                         |
> | 0.2834 | 0.6255 | 0.7910 | 0.4615 | 0.5120  | 0.4393| 1    | 96                         |
> | 0.2857 | 0.6279 | 0.7762 | 0.4641 | 0.5123  | 0.4411| 2    | 96                         |
> | 0.2816 | 0.6179 | 0.7675 | 0.4559 | 0.5045  | 0.4341| 1    | 128                        |
> | 0.2808 | 0.6183 | 0.7668 | 0.4563 | 0.5045  | 0.4341| 2    | 128                        |
> | 0.2814 | 0.6100 | 0.7525 | 0.4523 | 0.4986  | 0.4312| 1    | 196                        |
> | 0.2828 | 0.6057 | 0.7492 | 0.4506 | 0.4972  | 0.4308| 2    | 196                        |
> | 0.2746 | 0.5954 | 0.7405 | 0.4413 | 0.4885  | 0.4222| 1    | 256                        |
> | 0.2732 | 0.5953 | 0.7448 | 0.4409 | 0.4894  | 0.4221| 2    | 256                        |
>
>
> - These results reinforce our understanding of the ConvFormer model's behavior concerning these hyperparameters. The choice of N=2 blocks and D=64 hidden units seems to strike a balance, capturing the essential patterns in the dataset without unnecessarily increasing complexity.

---

> ### Author Response · Authors · 2023-11-18
> **Response to Reviewer DKuE [4/4]**
>
> ### **Reference**
>
> [1] Zhou, Kun, et al. "Filter-enhanced MLP is all you need for sequential recommendation." Proceedings of the ACM web conference 2022. 2022.
>
> [2] Kang, Wang-Cheng, and Julian McAuley. "Self-attentive sequential recommendation." 2018 IEEE international conference on data mining (ICDM). IEEE, 2018.
>
> [3] Sun, Fei, et al. "BERT4Rec: Sequential recommendation with bidirectional encoder representations from transformer." Proceedings of the 28th ACM international conference on information and knowledge management. 2019.
>
> [4] Li, Muyang, et al. "MLP4Rec: A Pure MLP Architecture for Sequential Recommendations." 31st International Joint Conference on Artificial Intelligence and the 25th European Conference on Artificial Intelligence (IJCAI-ECAI 2022). International Joint Conferences on Artificial Intelligence, 2022.

---

> ### Author Response · Authors · 2023-11-22
> **Thank you for your constructive feedback.**
>
> We sincerely thank you for your constructive feedback. The suggested experiments are quite interesting and effective to improve the quality of this work. We have provided the empirical evidence in author response, which facilitates to validate the efficacy of ConvFormer and corresponding criteria.
>
> As the deadline for our discussion is approaching, we anticipate any further technical advice or inquiries and we will make our best effort to handle them promptly and thoroughly.

---

> > ### Comment · Reviewer_DKuE · 2023-11-23
> > **Thanks for the reply!**
> >
> > Hi authors,
> >
> > Thanks for the detailed explaination and I have read all of them. I really appreciate you taking the time to answer my question. I will keep my rating but I do think this is an interesting and insightful paper. Nice work!

---

> ### Author Response · Authors · 2023-11-28
> **Thanks for your appreciation!**
>
> Dear Reviewer DKuE,
>
> Thank you for your follow-up comment and appreciation to this work. We would be very happy if our observations could offer useful inspirations in academic or industrial practices.
>
> Recently, **we came across a submission to ICLR, "ModernTCN: A Modern Pure Convolution Structure for General Time Series Analysis" [1], which we believe bears relevance to our work and could further reinforce the confidence in our findings.**
>
> - Overall, modernTCN [1] is featured by a modernTCN layer with applications to sequence modeling, closely aligning with our LighTCN block in terms of technical aspects. It focuses on the research problem:  `Can CNN outperform TransFormer in sequential modeling?` The community’s positive reception of "ModernTCN" (reflected in its scores of 8-8-8) could corroborate the practical efficacy of our LighTCN block.
> - **In comparison, our work’s innovation and contribution extend beyond the specific LighTCN block**. We delve into a critical reassessment of the current trend favoring simpler methods over Transformer-style approaches.
> In doing so, we not only investigated the research problem:  `Can CNN outperform TransFormer in sequential modeling?`, but also went further to explore `Are there common criteria for effective token-mixers in sequential modeling?` This exploration has led to the formulation of three pivotal criteria. Adhering to the criteria, simple token mixers (e.g., even an affine layer in Appendix F) can achieve leading performance. **The significance of our work, therefore, lies in broader implications for architectural methodology in sequential user modeling, offering key insights for future innovations, rather than proposing a specific CNN-like block with SOTA performance.**
>
> [1] "ModernTCN: A Modern Pure Convolution Structure for General Time Series Analysis." https://openreview.net/forum?id=vpJMJerXHU
>
> Given the technical similarities (albeit the difference of research problems) with [1], we are hopeful that the review and comments of [1]  will further solidify your support for our submission. Such support becomes particularly crucial in light of a contrasting emergent review we have encountered, to which we have responded comprehensively. In a field where parallel works coexist, such acknowledgment and support from active reviewers are invaluable. This is of course not a demanding, but a kind reinforcement of your confidence.
>
> With five days remaining for discussion, we are fully committed to engaging in any further technical dialogues you may have.

---

### Official Review · Reviewer_dYvo · 2023-11-25

**Soundness:** 2 fair
**Presentation:** 2 fair
**Contribution:** 1 poor
**Rating:** 3
**Confidence:** 4

**Summary:**

Despite the great success of transformer models in the research community of machine learning, this paper argues that such an architecture is not better than other simpler baselines (like MLP) on sequential user modeling tasks.  To develop optimal architectures, the authors identify three key factors for sequential models: sensitivity to the order of input sequences, lightweight model size, and large receptive field.  Based on these criteria, this paper proposes ConvFormer, which replaces self-attention modules with convolutional layers in the transformer models.  The authors demonstrate consistent performance gain on four sequential user modeling datasets with the proposed architectures.

**Strengths:**

1. The paper demonstrates consistent performance gains on most benchmark datasets

**Weaknesses:**

Overall, the proposed method is obsolete.  This paper does not follow recent progress in the machine learning community.  The followings are the weaknesses.

---

### Insufficient literature review

1. The formulation of permutation-variant (or order-sensitive) self-attention is outdated.  There are enormous ways to equip self-attention with the sensitivity to input sequence order.  Just to name few of them:

    * **Learnable absolute embedding**: Training data-efficient image transformers & distillation through attention, Touvron et al, ICML 2021
    * **Sinusoidal absolute embedding**: Attention is All You Need, Vaswani et al, NeuRIPS 2017.
    * **Rotary relative embedding**: RoFormer: Enhanced Transformer with Rotary Position Embedding, Su et al, arXiv 2021
    * **Learnable relative embedding**: Swin Transformer V2: Scaling Up Capacity and Resolution, Liu et al, CVPR 2022.

All these modelings have shown superior performance in NLP, CV, and Robotics.

2. Eqn (1) should cite `Vaswani et al.`, which is the seminal work for proposing the self-attention formulation

3. Eqn (2) should cite `Vasani et al.` and `Dosovitskiy et al` (An Image is Worth 16x16 Words: Transformers for Image Recognition at Scale).  Summing input tokens with position encodings is a common procedure in transformer models.

4. The author does not cite `Mathieu et al` (Fast Training of Convolutional Networks through FFTs), which proposes to accelerate convolutional operator using FFT in 2013.

---

### Odd formulations

5. The formulation of SAR-P seems not sensitive to the order of input sequence but the features of the input token $R[l]$

6. The formulation of window-masked self-attention is odd.  One key component in self-attention is to fix the numerical scale of aggregated features $S = A(RW^{(V)})$, where the summation of each row in $A$ is constrained to be $1$.  To apply window masking, a common practice is to set the entries, outside the window, in $A$ to $-\inf$ (see Sec 3.2.3 in `Vasani et al.`).

However, the formulation in Sec 3.2 of this paper does not guarantee scale invariance of feature aggregation.  In extreme cases, where $A$ has very small attention within the window, the scales of aggregated features become $0$.

7. The ablation of receptive field in Fig 1 is weird.  Existing works usually study the effect of receptive field with convolutional models `Liu et al.` (A ConvNet for the 2020s)

---

### Overclaims

8. This paper proposes to replace self-attention modules with convolutional layers.  The idea is not novel, as the final model becomes Residual Network `He et al.` (Deep Residual Learning for Image Recognition).  In fact, there are papers comparing transformer architectures with convolutional UNet for sequential modeling tasks, like `Chi et al.` (Diffusion Policy: Visuomotor Policy Learning via Action Diffusion).  Moreover, there are also works integrating unify both sides in a single model `Rombach et al.` (High-Resolution Image Synthesis with Latent Diffusion Models)

9. The idea of using convolution in sequential user modeling tasks is already explored in `Zhou et al., 2022`

**Questions:**

See above weaknesses

---

> ### Author Response · Authors · 2023-11-26
> **Response to dismissive emergent review by dYvo [1/2]**
>
> Thanks for your comment. Let me be straightforward: **you focus on correctable issues such as reference and discussable technical formulations, while missing the main contribution of this paper.**
>
> > The formulation of permutation-variant (or order-sensitive) self-attention is outdated.
> - **We never discuss the concept `order-sensitive self-attention`.** The core of Section 3.1 is the order **IN**sensitivity of vanilla self-attention models.
> - We focused on the order-sensitivity of **token-mixer**, rather than the role of positional embeddings (PEs). PEs do matter (and we also involved it) but their role in preserving order information is marginal, a perspective supported by extensive literature including TxzE's comment: `positional embeddings also count but their contribution is marginal`.
>
> > Eqn (1) should cite Vaswani et al., which is the seminal work for proposing the self-attention formulation
> - **`Vaswani et al.` were appropriately cited in the related work section.**
> For Eqn (1), our citation of `Kang & McAuley, 2018` is **intentional**, as it is the seminal work for implementing self-attention within the domain of sequential user modeling, **aligning with our aim** to contextualize self-attention within sequential user modeling in this section.
>
>
> > Eqn (2) should cite Vasani et al. and Dosovitskiy et al.
> - **We cited `Zhou et al., 2022`, a recent literature that used PE in Eqn (2) in our field.** Recognizing the extensive literature on learnable PEs in diverse fields like NLP and computer vision, we opted for **recent literature in our primary area with similar settings**. The foundational role of the vanilla Transformer by `Vasani et al.` is also acknowledged in our related work section.
>
> > The author does not cite Mathieu et al.
> - We cited `Oppenheim et al., 2001` instead, which is a seminal work for Fourier convolution. We agree to add `Mathieu et al.`` to reference, while still emphasizing the role of seminal work `Oppenheim et al., 2001`  (we believe you should support it according to your Q2).
>
>
> **In summary, our manuscript cites most references correctly and appropriately, prioritizing recent literature that is most pertinent to our primary area of study.**
>
> ---
>
> > The formulation of SAR-P seems not sensitive to the order of input sequence but the features of the input token
> - **SAR-P is sensitive to the order of input sequence**. We provided analysis in Appendix C.2.
> - The order sensitivity can be defined as follows: for any item in the input sequence, changing the order of other items alters the representation outputted by the token mixer. To illustrate this, consider the representation of the $i$-th item in the input sequence, denoted as $r_i \in \mathbb{R}^\mathrm{D}$.
> In SAR and its variants, the output representation of $r_i$ is given by $s_i=\sum a_{i,j}r_j$ (we omit the value mapping here for brevity. It is reasonable since the value mapping is conducted on each step isolatedly, without impact on order sensitivity).
> If we switch the order of two items, $j_1$ and $j_2$, the output representation changes accordingly: $s_i^\prime=\sum_{j=1,j\neq{j_1},j\neq j_2}^{\mathrm{L}}a_{i,j}r_j + a_{i,j_1}r_{j_2}+a_{i,j_2}r_{j_1}$, reflecting the model's order sensitivity. **In SAR**, $s_i$ does not exhibit order sensitivity since $s^\prime_i=s_i$. **In SAR-O**, $a_{i,j}$ are fixed w.r.t. positions, which makes $s^\prime_i \neq s_i$. **In SAR-P**, the only difference from SAR-O is that the weights $a_i$ are dependent on $r_i$: a_i=[a_{i,j}]_{j=1:L}=\mathrm{MLP}(r_i). **Since the generated weights are dependent on the position index $j$, $s{^\prime}_i\neq s_i$ holds and thus SAR-P is order-sensitive.**
> - If the statements above do not clarify your confusion, **could you elaborate on your perspective regarding SAR-P's order insensitivity?**
>
>
> > The formulation of window-masked self-attention is odd.
> - Our practical implementation adhered to the scale-invariant approach you mentioned, as evidenced in our code (`lines 242-247 in src/modules.py`). The manuscript initially used a simplified description for clarity; but this simplification ignored the role of scale. We will revise the statement accordingly.
>
> > The ablation of receptive field in Fig 1 is weird. Existing works usually study the effect of receptive field with convolutional models.
> - The criticism is not clear to us.  Our aim in Section 3 is to comprehensively ablate Transformer models to identify factors influencing performance, and thus we conduct experiments on Transformer. **Is it prohibitive to study the effect using different models with current works?**
>
> **In summary, the critism comes from potential misunderstandings or ambiguities in formulation and experimental design.** We have revised manuscript to clarify these points.

---

> > ### Comment · Reviewer_dYvo · 2023-11-26
> >
> > I would like to thank the authors for their response.  I would also like to point out that good research papers are often accessible to readers outside the research field and thus facilitate other research fields.  For example, the original self-attention paper (`Vaswani et al., 2017`) facilitates the recent progress in computer vision.  I would like to encourage the author to improve accessibility, especially for a model-architecture paper.
> >
> > I would like to also apologize if I missed any comment from other reviewers, since I didn't want to be biased by others for providing my first review of this paper.
> >
> > The followings are my responses.
> >
> > ---
> >
> > > >The formulation of permutation-variant (or order-sensitive) self-attention is outdated.
> > >
> > >    We never discuss the concept order-sensitive self-attention. The core of Section 3.1 is the order INsensitivity of vanilla self-attention models.
> > > We focused on the order-sensitivity of token-mixer, rather than the role of positional embeddings (PEs). PEs do matter (and we also involved it) but their role in preserving order information is marginal, a perspective supported by extensive literature including TxzE's comment: positional embeddings also count but their contribution is marginal.
> >
> > The author mentioned the word `order` 12 times, the `sensitive/sensitivity` 8 times in this section, which did discuss the concept of order sensitivity.  At least, based on the listed SAR-O/P/R, I thought the authors aimed to discuss variants of order-sensitive self-attention models in this section.
> >
> > After reading  `Kang & McAuley, 2018`, I just realized that the self-attention $A$ in Eqn (1) involves positional encodings (PE) and the corresponding performance gain is marginal.  However, the involvement of PE in $A$ is not mentioned clearly in Section 2 where the self-attention formulation is first brought up.  The author only mentioned one line at the bullet list of SAR-O, which is confusing.
> >
> > ----
> >
> > >>    Eqn (1) should cite Vaswani et al., which is the seminal work for proposing the self-attention formulation
> > >
> > >   Vaswani et al. were appropriately cited in the related work section. For Eqn (1), our citation of Kang & McAuley, 2018 is intentional, as it is the seminal work for implementing self-attention within the domain of sequential user modeling, aligning with our aim to contextualize self-attention within sequential user modeling in this section.
> >
> > My reasons are two-fold: 1. the formulation is exactly the same as that of `Vaswani et al` (Eqn (1) in the paper), and 2. `Vaswani et al` also tackled sequential modeling tasks (specifically NLP).  The sequential user modeling task predicts the probability of next item, while, `Vaswani et al` predicts the probability of the next token (Sec 3.4 in the paper).  `Vaswani et al` is closely related to Sec 2.
> >
> > ---
> >
> > >> Eqn (2) should cite Vasani et al. and Dosovitskiy et al.
> > >
> > > We cited Zhou et al., 2022, a recent literature that used PE in Eqn (2) in our field. Recognizing the extensive literature on learnable PEs in diverse fields like NLP and computer vision, we opted for recent literature in our primary area with similar settings. The foundational role of the vanilla Transformer by Vasani et al. is also acknowledged in our related work section.
> >
> > I (kind of) agree with the response.  However, `Vasani et al.` and `Dosovitskiy et al.` are seminal works in the machine learning community.  Citing them could increase accessibility of this paper.
> >
> > ---
> >
> > >> The author does not cite Mathieu et al.
> > >
> > > We cited Oppenheim et al., 2001 instead, which is a seminal work for Fourier convolution. We agree to add Mathieu et al.`` to reference, while still emphasizing the role of seminal work Oppenheim et al., 2001` (we believe you should support it according to your Q2).
> >
> > The main difference between these two is that `Mathieu et al` proposes to employ FFT to accelerate **deep** convolutional neural networks.  To the best of my knowledge is the first paper after the introduction of AlexNet, when the application of deep CNN became popular in the machine learning community.  I believe it is natural to cite `Mathieu et al` in this paper.

---

> > > ### Comment · Reviewer_dYvo · 2023-11-26
> > >
> > > >> The formulation of SAR-P seems not sensitive to the order of input sequence but the features of the input token
> > > >
> > > > SAR-P is sensitive to the order of input sequence. We provided analysis in Appendix C.2.
> > > >
> > > > The order sensitivity can be defined as follows: for any item in the input sequence, ...
> > > >
> > > > If the statements above do not clarify your confusion, could you elaborate on your perspective regarding SAR-P's order insensitivity?
> > >
> > > I checked the appendix, and my concern is addressed.
> > >
> > > ---
> > >
> > > >> The formulation of window-masked self-attention is odd.
> > > >
> > > > Our practical implementation adhered to the scale-invariant approach you mentioned, as evidenced in our code (lines 242-247 in src/modules.py). The manuscript initially used a simplified description for clarity; but this simplification ignored the role of scale. We will revise the statement accordingly.
> > >
> > > I'd encourage the author to correct the paper correspondingly.
> > >
> > > ---
> > >
> > > >> The ablation of receptive field in Fig 1 is weird. Existing works usually study the effect of receptive field with convolutional models.
> > > >
> > > > The criticism is not clear to us. Our aim in Section 3 is to comprehensively ablate Transformer models to identify factors influencing performance, and thus we conduct experiments on Transformer. Is it prohibitive to study the effect using different models with current works?
> > >
> > > My concern is that in vanilla SAR, the receptive field is natural to be as large as the whole input window.  Though the window size are often constrained via masking to preserve the auto-regressive property (see Section 3.2.3 in `Vaswani et al.`), the last language token still attend to every tokens in the input sequence.  It is thus odd to study the receptive field size via SAR.
> > >
> > > On the other hand, ConvNext (`Liu et al, 2022`) showed that CNN can be as effective as Transformers.  One trick is to increase the receptive field size (see Fig 2 in the paper).  It is thus natural to study the effect of receptive field size with CNN, especially, this paper aims to propose a convolutional architecture to replace SAR.

---

> > > > ### Comment · Reviewer_dYvo · 2023-11-26
> > > >
> > > > >> This paper proposes to replace self-attention modules with convolutional layers. The idea is not novel, as the final model becomes Residual Network He et al. (ResNet).
> > > > >
> > > > > The model significantly diverges from ResNet. (1) ResNet used a limited receptive field, which contradicts Criterion 2 of our study and is shown to be ineffective in Section 5.3.1. (2) ResNet used conventional convolution operator, which contradicts Criteria 3 and is shown to be ineffective in Section 5.3.2. (3) ResNet does not conform to the meta-former architecture, which is central to our investigation focused on the token mixer design within this framework.
> > > > >
> > > > > It's noteworthy that if researchers used ResNet-like model to beat advanced Transformers in their specific field and explored the rational behind with theoretical or empirical evidence, the work would be fairly valuable that should not be summarily dismissed. See [1] for reference.
> > > >
> > > > There are three differences between ResNet and this model: 1. ResNet does not use depth-wise convolution, 2. ResNet integrate an 1x1 Convolutional layer in each residual block, while, this method has two separate sub-blocks in each lightTCN block (NxD depth-wise conv and 1xD conv), and 3. the receptive field of this method covers the whole input window.  The scientific contribution of having these three differences is trivial.  For example, ConvNext (`Liu et al, 2022`) also proposes to adopt depth-wise convolution, yet, the paper also studies many other architectural designs in details, even including the training recipe.
> > > >
> > > > On the other hand, I'm not sure why conforming to the meta-former architecture is an important scientific question.  For me, it's just a choice of hyper-parameters where every layer share similar structures.
> > > >
> > > > Nonethelss, I appreciate and had little concern of the performance gain brought by this paper.  My low-score rating is mainly based on the low-quality presentation of this paper.
> > > >
> > > > ---
> > > >
> > > > >> The idea of using convolution in sequential user modeling tasks is already explored in Zhou et al., 2022
> > > > >
> > > > > Our contribution transcends a specific architecture. It re-evaluates Transformer-like architectures and identifies three criteria for effective token mixers in sequential user modeling. ConvFormer serves as a proof-of-concept, illustrating that even simplified models adhering to our proposed criteria can excel over more complex solutions. This concept is further corroborated by experiments with other simple models, such as an affine token mixer detailed in Appendix F.
> > > >
> > > > This paper does not mention that `Zhou et al., 2022` also adopts the similar idea of using convolutional layer.  Instead, in the introduction, this paper says *"This has led to instances where simpler approaches,
> > > > such as MLP-like modules, outperform the more complex self-attentive token mixers in Transformers (Zhou et al., 2022)."*  The paragraph misleads readers to appreciate the novelty of using convolutional layers in this paper, which is already proposed in `Zhou et al., 2022`.

---

> ### Author Response · Authors · 2023-11-26
> **Response to dismissive emergent review by dYvo [2/2]**
>
> > This paper proposes to replace self-attention modules with convolutional layers. The idea is not novel, as the final model becomes Residual Network He et al. (ResNet).
>
> - **The model significantly diverges from ResNet.** (1) ResNet used a limited receptive field, which contradicts Criterion 2 of our study and is shown to be ineffective in Section 5.3.1. (2) ResNet used conventional convolution operator, which contradicts Criteria 3 and is shown to be ineffective in Section 5.3.2. (3) ResNet does not conform to the meta-former architecture, which is central to our investigation focused on the token mixer design within this framework.
> - It's noteworthy that if researchers used ResNet-like model to beat advanced Transformers in their specific field and explored the rational behind with theoretical or empirical evidence, **the work would be fairly valuable that should not be summarily dismissed**. See [1] for reference.
>
> > The idea of using convolution in sequential user modeling tasks is already explored in Zhou et al., 2022
> - Our contribution transcends a specific architecture. It re-evaluates Transformer-like architectures and identifies three criteria for effective token mixers in sequential user modeling. ConvFormer serves as a **proof-of-concept**, illustrating that even simplified models adhering to our proposed criteria can excel over more complex solutions. This concept is further corroborated by experiments with **other simple models**, such as an affine token mixer detailed in Appendix F.
>
> **In summary, the criticisms appear to stem from a misunderstanding of the technical intricacies of ConvFormer and ResNet, as well as the innovation and contribution of this work.**
>
> [1] Wu, Haixu, et al. "Timesnet: Temporal 2d-variation modeling for general time series analysis." ICLR (2023).
>
> ---
> ### Conclusion
> We appreciate the unique perspective you bring to this review process. However, we must express that your comments, which **seem quite dismissive, stand in stark contrast to the feedback from other reviewers**.  The comments mainly involve **correctable / clarifiable issues** (although most of them have been clarified for many times in the manuscript) and **misunderstanding** of technical points. We have taken your comments seriously. We sincerely hope that you will consider our response with the same level of seriousness.

---

> ### Author Response · Authors · 2023-11-26
> **Thank you for your follow-up response [1/2]**
>
> Thanks for your prompt follow-up response. We noted that you checked the suggested reference and materials in preparing your response. In this duration, most concerns and divergences seem to be clarified. Below are my follow-up responses.
>
> ---
>
> > At least, based on the listed SAR-O/P/R, I thought the authors aimed to discuss variants of order-sensitive self-attention models in this section.
> - We hoped to clarify that `order-sensitive SAR variants` (in our work) are not equivalent to `order-sensitive self-attention` models (in your comment). They are different, since the suggested SAR variants are not always self-attention models that rely on item-to-item attention.
>
> > After reading Kang & McAuley, 2018, I just realized that the self-attention in Eqn (1) involves positional encodings (PE) and the corresponding performance gain is marginal. However, the involvement of PE in
>  is not mentioned clearly in Section 2 where the self-attention formulation is first brought up. The author only mentioned one line at the bullet list of SAR-O, which is confusing.
> - We formulated the implementation of PE in Section 4. Note the task of Section 2 (namely Problem Statement) is introducing the self-attention matrix $A$ in the sequential user modeling problem, which is the basis for describing the SAR variants in Section 3, rather than describing an end-to-end pipeline. So we moved the details including PE to Section 4. In section 3, we also highlighted that we used the same trick (residual link+LN, meta-former architecture, PE, etc.) with SASRec in the footnote.
>
> > My reasons are two-fold: 1. the formulation is exactly the same as that of Vaswani et al (Eqn (1) in the paper), and 2. Vaswani et al also tackled sequential modeling tasks (specifically NLP). The sequential user modeling task predicts the probability of next item, while, Vaswani et al predicts the probability of the next token (Sec 3.4 in the paper). Vaswani et al is closely related to Sec 2. I (kind of) agree with the response. However, Vasani et al. and Dosovitskiy et al. are seminal works in the machine learning community. Citing them could increase accessibility of this paper. The main difference between these two is that Mathieu et al proposes to employ FFT to accelerate deep convolutional neural networks. To the best of my knowledge is the first paper after the introduction of AlexNet, when the application of deep CNN became popular in the machine learning community. I believe it is natural to cite Mathieu et al in this paper.
>
> - In the initial submission, we have cited `Vasani et al.` and highlighted its role (although it should be a consensus) in the related work section. We agree that the suggested papers, e.g., `Vasani et al.`, `Mathieu et al.` and `Dosovitskiy et al.` are also seminal works and citing them could increase accessibility of this paper, and will cite them (or cite them more) in the corresponding places. Nevertheless, we do think such points are discussable as you demonstrated and thus do not suffice a critical critism for insufficient literature review.
>
> ---
>
> >>The formulation of SAR-P seems not sensitive to the order of input sequence but the features of the input token.
>
> > I checked the appendix, and my concern is addressed.
>
> - **Thanks for your positive feedback.**
>
> The formulation of window-masked self-attention is odd.
>
> >>The formulation of window-masked self-attention is odd.
>
> >I'd encourage the author to correct the paper correspondingly.
> - **It seems that we reach an agreement here.** We have revised it accordingly but the revision cannot be uploaded at this stage. The revised version is as follows.
>   - This is achieved by using a window mask $\Gamma(\mathrm{K})$, defined such that $\Gamma_{ij} = 1$ if $|i-j|\leq \mathrm{K}$ and $\mathrm{-inf}$ otherwise for $0\leq i, j \leq \mathrm{L}$. The attention matrix $\mathbf{A}$ in (1) is multiplied with $\Gamma(\mathrm{K})$ before calculating Softmax.
>
>
> > My concern is that in vanilla SAR, the receptive field is natural to be as large as the whole input window ... It is thus odd to study the receptive field size via SAR.
> - Section 3 **aims to** deconstruct self-attentive token mixer to investigate the key factors that improve or limit the performance of Transformer-style user model. In this way we can identify criteria for effective token mixers in sequential user modeling. **To this end, we ablated receptive field on SAR**. **Could you please elaborate on the specific inaccuracies here if applicable?**

---

> ### Author Response · Authors · 2023-11-26
> **Thank you for your follow-up response [2/2]**
>
> > The scientific contribution of having these three differences is trivial. For example, ConvNext (Liu et al, 2022) also proposes to adopt depth-wise convolution, yet, the paper also studies many other architectural designs in details, even including the training recipe.
>
> - Thanks for your clarification. Overall, ConvFormer does not degrade to ResNet (as opposed to your initial review) with some technical differences (in your updated response). While depth-wise convolution has been used in many fields like computer vision, **our work's pivotal contribution** lies in the **revisiting** of **Transformer-like architectures** in the field of **sequential user modeling**, identifying three criteria for effective token mixers. As a revisiting work, using simple and even trivial techniques (as we have claimed in the introduction section) to back up the proposed propositions is meaningful and even notable, **since it underscores the necessarity of such revisiting and the efficacy of the proposed propositions.**
> For instance, linear layer could beat Transformer in time series forecast [1]; the trival technical point and the diverse application of the linear layer do not weaken the significance or contribution of [1].
>
> > Instead, in the introduction, this paper says "This has led to instances where simpler approaches, such as MLP-like modules, outperform the more complex self-attentive token mixers in Transformers `(Zhou et al., 2022)`." The paragraph misleads readers to appreciate the novelty of using convolutional layers in this paper, which is already proposed in Zhou et al., 2022.
>
> - We recognized FMLP-Rec as an MLP since it  kept in line with the claim of `Zhou et al., 2022`, where the model is defined as an MLP model. The unintended misinterpretation of our wording is unexpected regrettable, **as our primary contribution transcends the proposition of a specific architecture**.  It is the revisiting of Transformer-like architectures in the field of sequential user modeling, identifying three criteria for effective token mixers. Other simple architectures that satisfy the proposed criteria, e.g., the affine layer in Appendix F, can also serve as proof-of-concept of the proposed criteria.
>
>
> [1] Zeng, Ailing, et al. "Are transformers effective for time series forecasting?." AAAI (2023).
>
> ---
>
> ### Conclusion
>
> We appreciate your follow-up response.
> We observe that most concerns seem to be clarified, and the remaining comments mainly involve correctable / clarifiable issues. We have taken your insights into serious consideration and look forward to your response.  **You are highly encouraged to open a new thread to summarize your remaining concerns for further discussion.** If applicable, you can revise your initial score based on your current perspective.

---

> > ### Comment · Reviewer_dYvo · 2023-11-28
> >
> > Thanks for the response.
> >
> > ---
> >
> > >> My concern is that in vanilla SAR, the receptive field is natural to be as large as the whole input window ... It is thus odd to study the receptive field size via SAR.
> > >
> > >    Section 3 aims to deconstruct self-attentive token mixer to investigate the key factors that improve or limit the performance of Transformer-style user model. In this way we can identify criteria for effective token mixers in sequential user modeling. To this end, we ablated receptive field on SAR. Could you please elaborate on the specific inaccuracies here if applicable?
> >
> > The main concern is that the conclusion derived from self-attention modules may not apply to convolutional layers, to verify it, one must conduct ablation study with convolutional layers.  Meanwhile, the paper proposes to replace self-attention with convolutional layers.  It is then natural to conduct ablation on convolution, not self-attention.
> >
> > ---
> >
> > >>    Instead, in the introduction, this paper says "This has led to instances where simpler approaches, such as MLP-like modules, outperform the more complex self-attentive token mixers in Transformers (Zhou et al., 2022)." The paragraph misleads readers to appreciate the novelty of using convolutional layers in this paper, which is already proposed in Zhou et al., 2022.
> > >
> > >    We recognized FMLP-Rec as an MLP since it kept in line with the claim of Zhou et al., 2022, where the model is defined as an MLP model. The unintended misinterpretation of our wording is unexpected regrettable, as our primary contribution transcends the proposition of a specific architecture. It is the revisiting of Transformer-like architectures in the field of sequential user modeling, identifying three criteria for effective token mixers. Other simple architectures that satisfy the proposed criteria, e.g., the affine layer in Appendix F, can also serve as proof-of-concept of the proposed criteria.
> >
> > Sec 4.2 in `Zhou et al., 2022` has pointed out that the filtering operation in Eqn 4, 5, and 6 is equivalent to convolutional operation.
> >
> > ---
> >
> > Overall, the presentation of this paper is not satisfying.  This paper does not cite previous works appropriately and not describe them clearly.  In addition, the method description is confusion and lacks certain details.  However, I believe that the authors have tried their best to address my concerns.  I would like to increase my rating from strong reject to reject.

---

> ### Author Response · Authors · 2023-11-28
> **Follow-up Response**
>
> **Thank you for summarizing your remaining concerns.** We understand your intention to provide **unbiased first** review and recognize the **variance** when reviewing the same or similar submissions [1]. However, we wish to emphasize that constructive reviews often evolve through dialogue, taking into account other reviews, author responses, and recent developments in the field.
>
> **Overall, there are two concerns remained.**
>
> ---
>
> > The main concern is that the conclusion derived from self-attention modules may not apply to convolutional layers, to verify it, one must conduct ablation study with convolutional layers. Meanwhile, the paper proposes to replace self-attention with convolutional layers. It is then natural to conduct ablation on convolution, not self-attention.
> - **We did involve ablation study with convolutional layers.** In Section 5.3.1, we noted `we vary the kernel size K to visualize its
> impact in Figure 4.` The kernel size K refers to the size of convolution kernel.
> - In fact, we conducted **ablations on both Transformer (Section 3) and ConvFormer (Section 5)** to verify the proposed criteria. Specifically, we started from the ablation of Transformer, which motivated the proposed criteria. Subsequently, we constructed ConvFormer, as a proof-of-concept of the proposed criteria, with superior overall performance. We further validated the criteria through ablation studies on ConvFormer.
>
> ---
>
> > Sec 4.2 in Zhou et al., 2022 has pointed out that the filtering operation in Eqn 4, 5, and 6 is equivalent to convolutional operation.
> - We wish to reiterate that the essence of our research **transcends the development of a specific architecture**. It is the revisiting of Transformer-like architectures in the field of sequential user modeling, identifying three criteria for effective token mixers. **Other simple architectures that satisfy the proposed criteria, e.g., the affine layer in Appendix F, can also serve as proof-of-concept of the proposed criteria (our core contribution in this field).**
> - It seems that `Zhou et al., 2022` can be classified into both MLPs and CNNs. In introduction, we initially respected their characterization as an all-MLP model based on their title and abstract: `an all-MLP model with learnable filters for sequential recommendation tasks`. We acknowledge that **this classification is open to discussion and is amendable** in our manuscript (respect the words by Zhou et al or the preference by yours). However, the core of our research is developing criteria for effective token mixing in our field, which transcends the development of a specific architecture.** Discussion over this particular classification should not overshadow the overall value of our work that leads to rejection.** We will follow your comment to revise the wording.
> ---
>
> ### Similar work in ICLR 2024
>
> - We find ModernTCN [1], a submission to ICLR 2024 in sequence modeling, which is featured by a modernTCN layer with applications to sequence modeling. The technical aspects of this work bear similarities to our LighTCN block. The community’s positive reception of "ModernTCN" (reflected in its scores of 8-8-8) corroborates the practical efficacy of our LighTCN block. It focuses on the research problem:  `Can CNN outperform TransFormer in sequential modeling?`.
> - **In comparison, our work’s contribution extend beyond the confines of CNN-like architecture**. It endeavors to critically reassess a prevailing trend in our field — the predominance of simple yet effective methods over Transformer-style methods in sequential modeling.
> In doing so, we not only addressed research problem:  `Can CNN outperform TransFormer in sequential modeling?`, but also went further to explore `Are there common criteria for effective token-mixers in sequential modeling?` This exploration has led to the formulation of three pivotal criteria. Adhering to the criteria, simple token mixers (e.g., even an affine layer in Appendix F) can achieve leading performance. The significance of our work, therefore, lies in its contribution to the broader architectural methodology in sequential user modeling, providing valuable insights for future architectural developments.
>
> [1] "ModernTCN: A Modern Pure Convolution Structure for General Time Series Analysis." https://openreview.net/forum?id=vpJMJerXHU
>
> ---
>
> The remaining concern seems a discussable wording selection in the introduction.
> Our research establishes three criteria for effective token mixing, verified on Transformer, ConvFormer, and even an affine layer. This core contribution, we hope to highlight, is not undermined by a discussable and amendable aspect of wording, and we will revise the wording according to your suggestion.
> We sincerely hope that our response is considered with an open and serious perspective. You are sincerely welcomed to summarize your remaining concerns in the next response.

---

> ### Author Response · Authors · 2023-11-30
> **A revision is uploaded for your reference**
>
> Dear Reviewer dYvo,
>
> We are pleased to know that the revision process for our manuscript is open today. We have carefully reviewed your summarized concerns, which primarily **focus on experiments with convolution models and the wording in the introduction**.
>
> As noted in our previous response, **we have indeed conducted experiments with convolution models**. In addition, we have taken into account your suggestion regarding the classification Zhou et al. While we respect the authors' preferred classification, we have also revised our introduction to **accommodate both perspectives**. Furthermore, in response to your initial comments regarding reference, we have **updated our selection of references**. We have also made revisions to our statement about the **window function**.
>
> All these revisions have been highlighted in **purple** in the **recently uploaded version** of our manuscript for your convenience.
>
> We sincerely hope that you will consider these revisions. We welcome and appreciate any further summarization of your remaining concerns in your next response. Your insights are valuable to us in enhancing the quality of our work.
>
> Thank you for your continued engagement and thoughtful feedback.

---

### Meta-Review · Area_Chair_uYRP · 2023-12-06

**Metareview:**

In this paper the authors introduce ConvFormer, "a simple yet effective" adaptation of the Transformer framework that satisfies three criteria for devising token-mixers in sequential user models: order sensitivity, a large receptive field, and a lightweight architecture.
Sequential user modeling is essential for building recommender systems, aiming to predict users’ subsequent preferences based on their historical behavior.  The authors note that despite the widespread success of the Transformer architecture in various domains,
we observe that its self-attentive token mixer is outperformed by simpler strategies in the realm of sequential user modeling.

Strengths:

-The paper demonstrates consistent performance gains on most benchmark datasets
-Identify the problem of self-attentive transformer when it was applied in user modeling; Good performance
-the topic of finding alternatives to self-attention mechanism is interesting and important, the paper can be viewed as a good overview of -
 the related literature
-the work contains a lot of empirical studies that allow greater granularity in analyzing the sequential learning process
-the idea of replacing self-attention with convolutional mechanisms has the potential to become a valuable contribution into the field


Weaknesses:
the description of the main experiments outlining key insights of the paper lack clarity
the design of the experiments and the methodology seem contradictory to the main claims of the paper
some straightforward baselines are missing from comparisons, which makes conclusion on superiority less convincing
No online real experiment Only the ID feature was considered


Weaknesses:
The first reviewer (dYvo) presented the strongest case for weakness in the paper:
(dYvo) The proposed method is obsolete. This paper does not follow recent progress in the machine learning community. The followings are the weaknesses: Insufficient literature review; Odd formulations; Overcliams
(DKuE) No online real experiment Only the ID feature was considered
(TxzE)   The description of the main experiments outlining key insights of the paper lack clarity
the design of the experiments and the methodology seem contradictory to the main claims of the paper
some straightforward baselines are missing from comparisons, which makes conclusion on superiority less convincing

**Justification For Why Not Higher Score:**

There were extensive weaknesses identified and the reviewers were not satisfied with the rebuttal.  I tend to agree with Reviewer dYvo that the described method is somewhat obsolete.

**Justification For Why Not Lower Score:**

There may be some merit to this paper but it needs to be rewritten.  The authors included such extensive new experiments that responding to the rebuttal involves nearly a re-review.

---

### Decision · Program_Chairs · 2024-01-16

Reject